# BinaryDM: Accurate Weight Binarization for Efficient Diffusion Models

**Xingyu Zheng[1], Xianglong Liu[✉1], Haotong Qin[2], Xudong Ma[1], Mingyuan Zhang[3],**
**Haojie Hao[1], Jiakai Wang[4], Zixiang Zhao[2,5], Jinyang Guo[1], Michele Magno[2]**
[1]Beihang University   [2]ETH Zürich   [3]Nanyang Technological University
[4]Zhongguancun Laboratory   [5]Xi'an Jiaotong University
{zhengxingyu,xlliu,macaronlin,haojiehao,jinyangguo}@buaa.edu.cn
{haotong.qin,michele.magno}@pbl.ee.ethz.ch mingyuan001@e.ntu.edu.sg
wangjk@zgclab.edu.cn zixiang.zhao@ethz.ch

## Abstract

With the advancement of diffusion models (DMs) and the substantially increased computational requirements, quantization emerges as a practical solution to obtain compact and efficient low-bit DMs. However, the highly discrete representation leads to severe accuracy degradation, hindering the quantization of diffusion models to ultra-low bit-widths. This paper proposes a novel weight binarization approach for DMs, namely **BinaryDM**, pushing binarized DMs to be accurate and efficient by improving the representation and optimization. From the representation perspective, we present an *Evolvable-Basis Binarizer* (EBB) to enable a smooth evolution of DMs from full-precision to accurately binarized. EBB enhances information representation in the initial stage through the flexible combination of multiple binary bases and applies regularization to evolve into efficient single-basis binarization. The evolution only occurs in the head and tail of the DM architecture to retain the stability of training. From the optimization perspective, a *Low-rank Representation Mimicking* (LRM) is applied to assist the optimization of binarized DMs. The LRM mimics the representations of full-precision DMs in low-rank space, alleviating the direction ambiguity of the optimization process caused by fine-grained alignment. Comprehensive experiments demonstrate that BinaryDM achieves significant accuracy and efficiency gains compared to SOTA quantization methods of DMs under ultra-low bit-widths. With 1-bit weight and 4-bit activation (W1A4), BinaryDM achieves as low as 7.74 FID and saves the performance from collapse (baseline FID 10.87). As the first binarization method for diffusion models, W1A4 BinaryDM achieves impressive $15.2\times$ OPs and $29.2\times$ model size savings, showcasing its substantial potential for edge deployment.

## 1 Introduction

Diffusion models (DMs) (Ho et al., 2020; Song & Ermon, 2019) have shown excellent capabilities in generation tasks in various fields, such as image (Ho et al., 2020; Song & Ermon, 2019; Song et al., 2020b), vision (Mei & Patel, 2023; Ho et al., 2022), and speech (Mittal et al., 2021; Popov et al., 2021; Jeong et al., 2021). DMs have become one of the most popular generative model paradigms with significant quality and diversity advantages. DMs generate data through the iterative noise estimates, while up to 1000 iterative steps slow the inference process and rely on expensive hardware resources. Although some proposed methods can effectively reduce the number of iterations to dozens of times (Song et al., 2020a; San-Roman et al., 2021; Nichol & Dhariwal, 2021; Bao et al., 2022), the complex neural network of DMs also results in a large number of floating point calculations and memory usage in each step, which hinders the efficient deployment and inference on edge. Therefore, the compression of DMs has been widely studied as a practical technology to accelerate the iterative process and reduce the inference cost, including quantization (Li et al., 2023b; Shang et al., 2023), distillation (Salimans & Ho, 2022; Luo, 2023; Meng et al., 2023), pruning (Fang et al., 2023), *etc*.

Low-bit quantization emerges as a practical approach to compress deep learning models by reducing the bit-width of parameters (Yang et al., 2019; Gholami et al., 2022; Qin et al., 2024), and also has

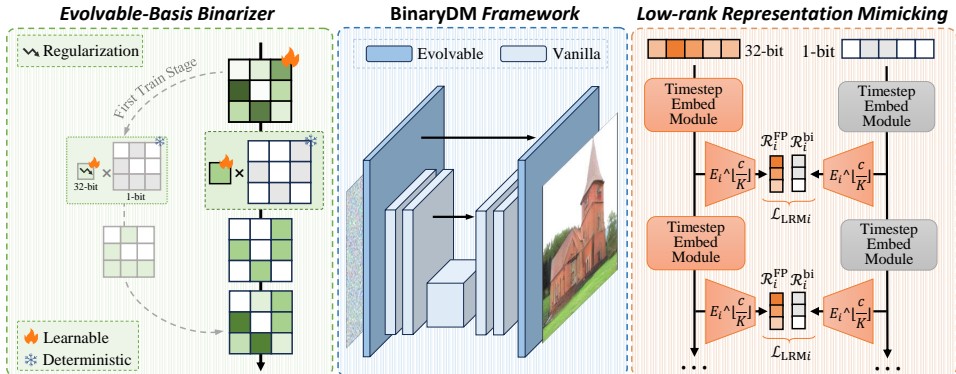

Figure 1: Overview of BinaryDM, consisting of Evolvable-Basis Binarizer to enhance information representation and Low-rank Representation Mimicking to improve optimization direction.

satisfactory generality to various network architectures. Thus, with quantization, DMs can enjoy the compression and acceleration brought by fixed-point parameters and computation in inference (Li et al., 2023b;a; Shang et al., 2023). The 1-bit quantization, namely binarization, allows the binarized model to enjoy compact 1-bit parameters and efficient computation (Liu et al., 2020; Xu et al., 2021b;a). With the most aggressive bit-width, 1-bit weights can lead to up to $32\times$ size reduction and replace expensive floating-point multiplications with addition constructions during inference, thus saving resources significantly (Rastegari et al., 2016; Frantar et al., 2023).

However, binarized DMs suffer significant performance degradation compared to their full-precision counterparts. The performance decline primarily arises from two aspects: **First**, weight binarization severely restricts the feature extraction capability of diffusion models, causing significant damage to information in critical representations of generative models. Though several weight binarization methods strive to mitigate binarization errors and enrich representations by floating-point scaling factors (Rastegari et al., 2016; Liu et al., 2020; Qin et al., 2023b; Zhang et al., 2024), the number of candidate values for each weight still drops from $2^{32}$ to $2^1$. The limited information-represent capacity of binarized filters leads to severe loss when compressing from full-precision initialization to 1-bit binarization. This fact causes catastrophic consequences for DMs that highly require representation capacity. **Second**, introducing discrete binarization functions in DMs poses a significant hurdle to stable convergence. Existing quantization-aware training methods for DMs usually employ direct output-based supervision (Li et al., 2023b; He et al., 2023). Binarization introduces significant errors in forward parameters and backward gradients, leading to disruptions in the optimization direction (Courbariaux et al., 2016). Learning the fine-grained details embedded in the synthetic features can contribute to the overall optimization process of binarized DMs. Unfortunately, the disruptive influence of extreme discretization becomes pronounced in this context, rendering the convergence vulnerable to disturbances and, in some cases, seemingly unattainable.

In this paper, we propose **BinaryDM** to push the weights of diffusion models toward binarization. The proposed method pushes the weights of diffusion models toward accurate and efficient binarization, considering the representation and computation properties. BinaryDM applies quantization-aware training to binarized DMs accurately for efficient inference, which takes the representation and computation properties of diffusion models into account and is composed of two novel techniques: *From the representation perspective*, we present an Evolvable-Basis Binarizer (EBB) to recover the representations generated by the binarized DM. EBB first applies dual sets of binary bases with learnable scalars to significantly enhance the feature extraction capability of the initial binarized weights, then evolves the high-order bases to the single-basis form guided by regularization loss. It is selectively applied only to key parameter locations of the DM architecture to reduce unnecessary evolution processes, easing the training burden and making the evolution smoother. *From the optimization perspective*, a Low-rank Representation Mimicking (LRM) is incorporated to enhance the binarization-aware optimization of diffusion models. LRM projects binarized and full-precision representations to low-rank, enabling the optimization of binarized DM to focus on the principal direction and mitigate direction ambiguity caused by the representation complexity of generation.

Comprehensive experiments show that our proposed BinaryDM has significant accuracy and efficiency gains compared to DMs binarized by existing SOTA binarization and low-bit quantization methods.

Our BinaryDM can consistently outperform the baseline on DDIM and LDM with binary weight, especially with ultra-low bit-width activation. For example, on CIFAR-10 32×32 DDIM, the precision metric of BinaryDM even exceeds the baseline by 9.46% (baseline 5.92 *vs.* BinaryDM 6.48) with 1-bit weight and 4-bit activation (W1A4), saving the binarized DM from collapse. BinaryDM even outperforms the higher bit-width SOTA quantization methods of DM. For LDM-8 on LSUN-Churches 256×256, W1A4 BinaryDM exceeds W4A4 EfficientDM in the FID metric by 4.43. As the first binarization method for DMs, BinaryDM yields impressive 15.2× and 29.2× savings on OPs and model size, demonstrating the vast advantages and potential for deploying the DM on edge.

## 2 RELATED WORK

**Diffusion models** (DMs) demonstrate outstanding performance across a diverse range of tasks (Ho et al., 2020; Song & Ermon, 2019; Song et al., 2020b; Niu et al., 2020; Mittal et al., 2021; Popov et al., 2021; Jeong et al., 2021; Peebles & Xie, 2023). However, their slow generation process presents a significant challenge to widespread implementation. Substantial research has focused on reducing the number of time steps to expedite the generation process (Watson et al., 2021; Chen et al., 2020; Song & Ermon, 2019; Song et al., 2020b; Feng et al., 2024b). Despite the reduction in time steps, the noise estimation network of DMs still demand expensive computation and memory for each step.

**Quantization and binarization** are explored widely as popular compression techniques (Nagel et al., 2020; Lin et al., 2021). These methods involve quantizing the full-precision parameters to lower bit-width (*e.g.*, 1-8 bit). By converting floating-point weights and activations into quantized values, the model size can be significantly reduced. This size reduction decreases computational complexity and substantially improves inference speed, memory usage, and energy consumption savings (Shang et al., 2023; Li et al., 2023a). One notable technique, quantization-aware training (Gholami et al., 2022; Qin et al., 2020; Yang et al., 2019), involves compressing DMs within a training/fine-tuning pipeline to update parameters (Li et al., 2023b; He et al., 2023). Despite these advancements, achieving 1-bit quantization for the weights of DMs remains a formidable challenge. This underscores the need for further research to unlock the potential benefits of 1-bit binarization in DMs. Appendix A presents more details about related works.

## 3 BINARYDM

### 3.1 PRELIMINARIES

In the forward process of diffusion models, Gaussian noise is added to data $\boldsymbol{x}_0 \sim q(\boldsymbol{x})$ in $T$ times via a schedule $\beta_t$ controlling noise strength, the process can be expressed as

$$q\left(\boldsymbol{x}_t \mid \boldsymbol{x}_{t-1}\right) = \mathcal{N}\left(\boldsymbol{x}_t; \sqrt{1-\beta_t}\boldsymbol{x}_{t-1}, \beta_t \boldsymbol{I}\right), \tag{1}$$

where $\boldsymbol{x}_t \in \{\boldsymbol{x}_1, \cdots, \boldsymbol{x}_T\}$ denote the noisy samples at $t$-th step. The reverse process aims to generate samples by removing noise, approximating the unavailable conditional distribution $q\left(\boldsymbol{x}_{t-1} \mid \boldsymbol{x}_t\right)$ with learned distributions $p_\theta\left(\boldsymbol{x}_{t-1} \mid \boldsymbol{x}_t\right)$, which can be expressed as

$$p_\theta\left(\boldsymbol{x}_{t-1} \mid \boldsymbol{x}_t\right) = \mathcal{N}\left(\boldsymbol{x}_{t-1}; \tilde{\boldsymbol{\mu}}_\theta\left(\boldsymbol{x}_t, t\right), \tilde{\beta}_t \boldsymbol{I}\right). \tag{2}$$

The mean $\tilde{\boldsymbol{\mu}}_\theta\left(\boldsymbol{x}_t, t\right)$ and variance $\tilde{\beta}_t$ could be derived using the reparameterization (Ho et al., 2020):

$$\tilde{\boldsymbol{\mu}}_\theta\left(\boldsymbol{x}_t, t\right) = \frac{1}{\sqrt{\alpha_t}}\left(\boldsymbol{x}_t - \frac{1-\alpha_t}{\sqrt{1-\bar{\alpha}_t}}\boldsymbol{\epsilon}_\theta\left(\boldsymbol{x}_t, t\right)\right), \qquad \tilde{\beta}_t = \frac{1-\bar{\alpha}_{t-1}}{1-\bar{\alpha}_t} \cdot \beta_t, \tag{3}$$

where $\alpha_t = 1 - \beta_t, \bar{\alpha}_t = \prod_{i=1}^t \alpha_i$, and $\boldsymbol{\epsilon}_\theta$ denotes a function approximation with the learnable parameter $\theta$, which predicts $\boldsymbol{\epsilon}$ from $\boldsymbol{x}_t$. The U-Net with spatial transformer layers is applied as the architecture of the noise estimation network in common practices. For the training of DMs, a simplified variant of the variational lower bound is usually applied as the loss function to achieve high sample quality, which can be expressed as

$$\mathcal{L}_{\text{simple}} = \mathbb{E}_{t, \boldsymbol{x}_0, \boldsymbol{\epsilon}_t}\left[\left\|\boldsymbol{\epsilon}_t - \boldsymbol{\epsilon}_\theta\left(\sqrt{\bar{\alpha}_t}\boldsymbol{x}_0 + \sqrt{1-\bar{\alpha}_t}\boldsymbol{\epsilon}_t, t\right)\right\|^2\right]. \tag{4}$$

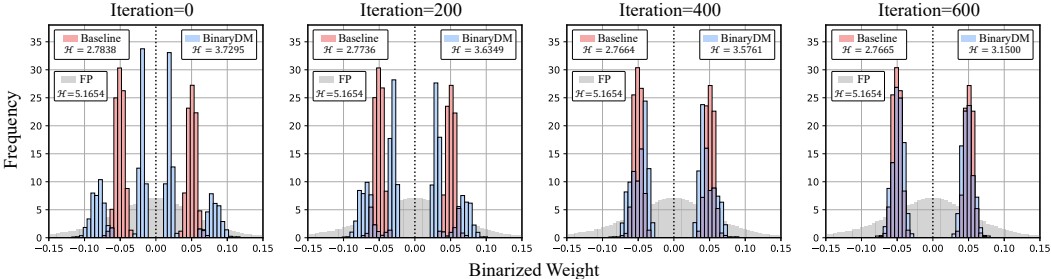

Figure 2: Comparison of binarized weights(channel-wise) for a convolutional layer. EBB possesses a broader representation range at the early stage and then gradually transitions to a single-basis state, while the quantitative information entropy $\mathcal{H}$ further illustrates its enhanced representation capacity.

The binarization and quantization compress and accelerate the noise estimation model by discretizing weights and activations to low bit-width. In the baseline of the binarized diffusion model, the weight $\boldsymbol{w} \in \theta$ is binarized to 1-bit by $\boldsymbol{w}^{\text{bi}} = \sigma \operatorname{sign}(\boldsymbol{w})$ (Rastegari et al., 2016; Courbariaux et al., 2016), where sign function confine $\boldsymbol{w}$ to +1 or -1 with 0 thresholds, $\boldsymbol{w}^{\text{bi}} \in \theta^{\text{bi}}$ denotes the binarized weight, and $\theta^{\text{bi}}$ denotes the binarized noise estimation network. $\sigma$ is the floating-point scalar, which is initialized as $\frac{\|\boldsymbol{w}\|}{n}$ ($n$ denotes the number of weight elements) and learnable during training process following (Rastegari et al., 2016; Liu et al., 2020). The activation is quantized by the LSQ quantizer (Esser et al., 2019). With the $32\times$ compressed weight, the computation of noise estimation can also be replaced with integer additions, achieving significant compression and acceleration.

## 3.2 Evolvable-Basis Binarizer

In the current baseline, weights are quantized to 1-bit values to economize on storage and computation during inference, and activations can be quantized to integers. However, the extensive discretization of weights to binary in DMs results in a notable deterioration of the generated representations. The bit-width of each weight element is limited to the original $\frac{1}{32}$, significantly restricting the information-carrying capacity of DMs. Previous works present a straightforward approach that enhances binarized parameters via higher-order residual bases (Li et al., 2017; Huang et al., 2024a; Chen et al., 2024a) have achieved significant success in terms of accuracy, but the introduced additional bases result in substantial additional hardware overhead, making them unsuitable for practical deployment on existing hardware architectures. While these methods do not achieve full binarization, they significantly help the model approach full-precision performance.

These findings led us to consider the significance of higher-order residual binarization for DMs, as it notably enhances the information space and improves representational capacity. To utilize the representation capability of high-order bases while avoiding redundant costs during inference, we sought to use residual binarized structures as transitional structures and evolve during training. This would allow fully binarized DMs to start from a more favorable initial state, resulting in a smoother optimization process and better final outcomes.

We propose the Evolvable-Basis Binarizer (EBB) to address the adaptation challenges binarized DMs face during the early optimization stages due to structural limitations. EBB is implemented in two stages during training. The first stage uses higher-order residual multi-basis with regularization penalties, then transitions into the second stage with simple single-basis binary weights.

**Learnable Multi-Basis**. In the forward propagation of the first stage, EBB is defined as

$$\boldsymbol{w}^{\text{bi}}_{\text{EBB}} = \sigma_{\text{I}} \operatorname{sign}(\boldsymbol{w}) + \sigma_{\text{II}} \operatorname{sign}(\boldsymbol{w} - \sigma_1 \operatorname{sign}(\boldsymbol{w})), \tag{5}$$

where the $\sigma_{\text{I}}$ and $\sigma_{\text{II}}$ are learnable scalars which are initialized as $\sigma_{\text{I}}^0 = \frac{\|\boldsymbol{w}\|}{n}$ and $\sigma_{\text{II}}^0 = \frac{\|\boldsymbol{w} - \sigma_1 \operatorname{sign}(\boldsymbol{w})\|}{n}$, respectively, $\|\cdot\|$ denotes the $\ell2$-normalization. The inference of layer binarized by EBB involves the computation of multiple bases. For instance, the convolution in binarized DM is

$$o = \boldsymbol{a} \times \boldsymbol{w}^{\text{bi}}_{\text{EBB}} = \sigma_{\text{I}} (\boldsymbol{a} \otimes \operatorname{sign}(\boldsymbol{w})) + \sigma_{\text{II}} (\boldsymbol{a} \otimes \operatorname{sign}(\boldsymbol{w} - \sigma_1 \operatorname{sign}(\boldsymbol{w}))), \tag{6}$$

where $\boldsymbol{a}$ denotes the activation, and $\times$ and $\otimes$ denote the convolution consisting of multiplication and addition instructions (Rastegari et al., 2016; Hubara et al., 2016), respectively.

In the backward propagation of EBB, the gradient of the learnable scalars is calculated as follows:

$$\frac{\partial \boldsymbol{w}_{\text{EBB}}^{\text{bi}}}{\partial \sigma_{\text{I}}} = \begin{cases} \text{sign}\,(\boldsymbol{w})\,(1 - \sigma_{\text{II}}\,\text{sign}\,(\boldsymbol{w})), & \text{if } \sigma_{\text{I}}\,\text{sign}\,(\boldsymbol{w}) \in (\boldsymbol{w} - 1, \boldsymbol{w} + 1), \\ \text{sign}\,(\boldsymbol{w}), & \text{otherwise}, \end{cases} \tag{7}$$

$$\frac{\partial \boldsymbol{w}_{\text{EBB}}^{\text{bi}}}{\partial \sigma_{\text{II}}} = \text{sign}\,(\boldsymbol{w} - \sigma_1\,\text{sign}\,(\boldsymbol{w})), \tag{8}$$

where the Straight Through Estimator (STE) is applied to approximate the $\text{sign}$ function during backward. With the binary basis with different learnable scalars, the representation capability of quantized weights can be significantly enhanced. The residual initialization makes the optimization of binarized DM start from an error-minimizing state. As presented in Figure 2, at the initialization at iteration-0, EBB exhibits significantly higher information entropy and a richer representational space. With EBB, the representation of weights is considerably more diversified than the binarized DM baseline, providing a more favorable initialization state for optimization.

**Transition Strategy**. We adopt a two-stage training process with a regularization strategy, allowing the DM to transition from an initial multi-basis structure to full binarization. In the first stage, regularization loss is applied to the higher-order learnable scaling factors, encouraging them to approach zero:

$$\mathcal{L}_{\text{EBB}} = \tau \frac{1}{N} \sum_{i=1}^{N} \sigma_{\text{II}}^{i}, \tag{9}$$

where $N$ denotes the number of basic layers (e.g., convolutional, linear) in the noise estimation network of DMs, and $\tau$ are hyperparameter coefficients used to balance the loss terms, typically set to 9e-2.

In the second stage, all higher-order terms are removed, and the forward propagation is simplified to:

$$\boldsymbol{w}^{\text{bi}} = \sigma_{\text{I}}\,\text{sign}\,(\boldsymbol{w}). \tag{10}$$

Through regularization penalties, EBB can smoothly evolve from an initially more information-rich residual state to a single-basis state suitable for inference. As shown by the evolution process in Figure 2, the dequantized weights of EBB gradually converge to a bimodal distribution consistent with full binarization as iterations progress. However, EBB consistently retains more information throughout the process, making the overall optimization of the binary DM easier.

**Location Selection**. In our BinaryDM, the proposed EBB is partially applied to crucial and parameter-sparse locations of the diffusion models while retaining concise vanilla binarization at other locations to reduce unnecessary evolution processes and the associated training overhead. Specifically, we apply EBB where the feature scale is greater or equal to $\frac{1}{2}$ input scale, *i.e.*, the first and last six layers with only the 15% of whole parameters in the noise estimation network of BinaryDM. In contrast, other layers keep consistent with the binarized DM baseline with vanilla binarizers. On the one hand, applying EBB to these key parameter locations within DM architectures significantly enhances the information processing capacity of binarized DMs in the early stages of optimization, leading to a better overall learning process. On the other hand, using a vanilla binarizer for intermediate layers, which contain the most parameters but are less sensitive to quantization loss, reduces the instability caused by switching between stages for unimportant components and lowers the training overhead.

### 3.3 Low-rank Representation Mimicking

In the quantization-aware training of DMs, the discretization of parameter space caused by weight binarization and activation quantization function and the inaccurate gradient approximation involved in the derivation process bring difficulties to the stable convergence of binarized DM. Since having almost the same architecture, the original full-precision DM can be regarded as an oracle of the binarized one. Therefore, an intuitive approach is to assist the training of binarized DMs by mimicking the representation of full-precision replicas. During training, aligning outputs and intermediate representations of binarized DMs with full-precision counterparts can provide additional supervision, accelerating the convergence of quantized DMs significantly.

However, there are issues directly aligning the intermediate representations of binarized and full-precision DMs during optimization. Firstly, fine-grained alignment of high-dimensional representation leads to a blurry optimization direction for DMs, especially when mimicking the intermediate

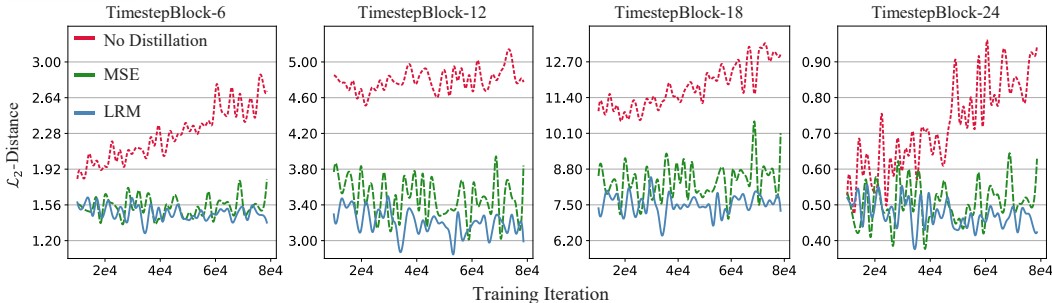

Figure 3: The impact of different distillation loss functions on the output features of each block in both full-precision DM and binary DM, measured by the $\mathcal{L}_2$ distance. Our proposed LRM enables the binarized DM to have the best information-mimicking capability.

features is introduced. Secondly, compared to the full-precision DM, the intermediate features in the binarized one are derived from a discrete latent space since the discretization of parameters makes it difficult to mimic the full-precision DM directly.

Therefore, we propose Low-rank Representation Mimicking (LRM) to efficiently optimize the BinaryDM by mimicking full-precision representations in a low-rank space. We group the full-precision DM $\theta^{\text{FP}}$ based on the timestep embedding modules composed of residual convolution and transformer blocks. The intermediate representation can be denoted as $\hat{\varepsilon}_{\theta_i}^{\text{FP}}(\boldsymbol{x}_t, t) \in \mathbb{R}^{h \times w \times c}$. We use principal component analysis (PCA) to project representations to low-rank space. The covariance matrix for representations of the full-precision DM is

$$C_i = \frac{1}{(h \times w)^2} \hat{\varepsilon}_{\theta_i}^{\text{FP}}(\boldsymbol{x}_t, t) \hat{\varepsilon}_{\theta_i}^{\text{FP}T}(\boldsymbol{x}_t, t), \tag{11}$$

where $\theta_i$ represents the composition of the top $i$ modules. The eigenvector matrix $E_i \in \mathbb{R}^{c \times c}$ is

$$E_i^T C_i E_i = \Lambda_i, \tag{12}$$

where $\Lambda_i$ is the diagonal matrix of eigenvalues of $C_i$, arranged in descending order. We take the matrix composed of the first $\lceil \frac{c}{K} \rceil$ column eigenvectors of $E_i$ as the transformation matrix, denoted as $E_i^{\lceil \frac{c}{K} \rceil}$, where $\lceil \cdot \rceil$ denotes the round function and $K$ denotes to the reduction times of dimension. We use $E_i^{\lceil \frac{c}{K} \rceil}$ to project the intermediate representation of both full-precision and binarized:

$$\boldsymbol{\mathcal{R}}_i^{\text{FP}}(\boldsymbol{x}_t, t) = \hat{\varepsilon}_{\theta_i}^{\text{FP}}(\boldsymbol{x}_t, t) E_i^{\lceil \frac{c}{K} \rceil}, \quad \boldsymbol{\mathcal{R}}_i^{\text{bi}}(\boldsymbol{x}_t, t) = \hat{\varepsilon}_{\theta_i^{\text{bi}}}^{\text{bi}}(\boldsymbol{x}_t, t) E_i^{\lceil \frac{c}{K} \rceil}, \tag{13}$$

where $\hat{\varepsilon}_{\theta_i}^{\text{bi}}(\boldsymbol{x}_t, t)$ denotes the intermediate representation of the $i$-th layer in the DM with binarized parameters $\theta^{\text{bi}}$, and $\boldsymbol{\mathcal{R}}_i^{\text{FP}}(\boldsymbol{x}_t, t)$ and $\boldsymbol{\mathcal{R}}_i^{\text{bi}}(\boldsymbol{x}_t, t)$ denote the low-rank representations of full-precision and binarized DMs, respectively, with the same shape $h \times w \times \lceil \frac{c}{K} \rceil$. The $K$ empirically defaults as 4 and is detailed ablated in Appendix B.2.

We then leverage the obtained low-rank representation to drive the binarized DM to learn the full-precision counterpart. We construct a mean squared error (MSE) loss between the $i$-th module of low-rank representations between full-precision and binarized DMs:

$$\mathcal{L}_{\text{LRM}i} = \left\| \boldsymbol{\mathcal{R}}_i^{\text{FP}} - \boldsymbol{\mathcal{R}}_i^{\text{bi}} \right\|. \tag{14}$$

The total loss function is composed of Eq.4, Eq.9 and Eq.14:

$$\mathcal{L}_{\text{total}} = \mathcal{L}_{\text{simple}} + \mathcal{L}_{\text{EBB}} + \lambda \frac{1}{M} \sum_{i=1}^{M} \mathcal{L}_{\text{LRM}i}, \tag{15}$$

where $M$ denotes the number of timestep embedding modules in the noise estimation network of DMs, and $\lambda$ is a hyperparameter coefficient to balance the loss terms, typically set to 1e-4.

Since the computation cost of obtaining the transformation matrix $E_i^{\lceil \frac{c}{K} \rceil}$ in LRM is significantly expensive, we compute the matrix by the first batch of input and keep it fixed during the training

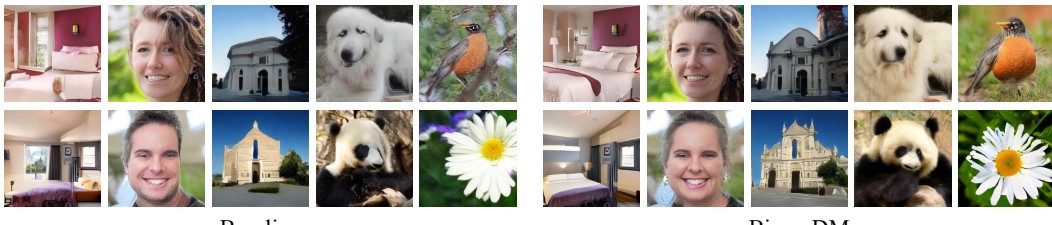

Baseline                                    BinaryDM

Figure 4: Visualization of samples generated by the binarized DM baseline and W1A4 BinaryDM.

Table 1: Comparison for unconditional generation on CIFAR-10 $32 \times 32$ by DDIM with 100 steps

| Method | #Bits | IS↑ | FID↓ | sFID↓ | Prec.↑ |
|---|---|---|---|---|---|
| FP | 32/32 | 8.90 | 5.54 | 4.64 | 67.92 |
| LSQ | 2/32 | 8.17 | 18.56 | 8.30 | 59.22 |
| Baseline | 1/32 | 7.84 | 22.59 | 6.83 | 60.23 |
| **BinaryDM** | 1/32 | **8.28** | **11.92** | **5.42** | **61.84** |
| LSQ | 2/8 | 7.64 | 29.66 | 30.63 | 58.76 |
| Baseline | 1/8 | 7.94 | 20.25 | 9.38 | 59.42 |
| **BinaryDM** | 1/8 | **8.47** | **11.21** | **5.49** | **62.65** |
| LSQ | 2/4 | 4.04 | 137.75 | 43.68 | 40.74 |
| Baseline | 1/4 | 5.92 | 100.17 | 51.06 | 36.46 |
| **BinaryDM** | 1/4 | **6.48** | **87.77** | **51.73** | **37.25** |

process. The fixed mapping between representations is also beneficial to the optimization of binarized DMs from a stability perspective, as updates to the transformation matrix could significantly alter the direction of binary optimization, which would be disastrous for DMs with high demands for representational capacity and optimization stability.

LRM enables binarized DMs to mimic the representation of full-precision counterparts, improving the optimization process by introducing additional supervision. As shown in Fig 3, LRM effectively brings the local block closer to the full-precision block. Furthermore, by applying low-rank projections based on the principal components from full-precision representations before representation mimicking, the binarized DM can be optimized along clear and stable directions, accelerating the convergence of the model. Furthermore, binarized and full-precision DMs have entirely consistent architectures, making representation mimicking between them natural.

## 4 EXPERIMENT

We conduct experiments on various datasets, including CIFAR-10 $32 \times 32$ (Krizhevsky et al., 2009), LSUN-Bedrooms $256 \times 256$ (Yu et al., 2015), LSUN-Churches $256 \times 256$ (Yu et al., 2015), FFHQ $256 \times 256$ (Karras et al., 2019) and ImageNet $256 \times 256$ (Deng et al., 2009), for both unconditional and conditional image generation tasks over DDIM and LDM. The evaluation metrics used in our study encompass Inception Score (IS), Fréchet Inception Distance (FID) (Heusel et al., 2017), Sliding Fréchet Inception Distance (sFID) (Salimans et al., 2016), and Precision-and-Recall. We implement and evaluate the DMs binarized by our BinaryDM and the baseline presented in Section 3.1, where LSQ (Esser et al., 2019) is employed uniformly as activations quantizers. Several SOTA quantization methods for DMs with 2∼8 bits weights are also considered (He et al., 2023; Li et al., 2023a;b; So et al., 2024). Detailed settings are presented in Appendix B.1.

### 4.1 MAIN RESULTS

**Unconditional Generation.** We first conduct experiments on the CIFAR-10 dataset. As shown in Table 1, the binarized DM baseline suffers a severe breakdown in this low-resolution scenario, while our method significantly recovers the performance. Under the W1A4 bit-width, BinaryDM surpasses

Table 2: Results for LDM on multiple datasets in unconditional generation by DDIM with 100 steps.

| Model | Dataset | Method | #Bits | Size(MB) | FID↓ | sFID↓ | Precision↑ | Recall↑ |
|---|---|---|---|---|---|---|---|---|
| LDM-4 | LSUN-Bedrooms 256 × 256 | FP | 32/32 | 1045.4 | 3.09 | 7.08 | 65.82 | 45.36 |
| | | LSQ | 2/32 | 69.8 | 7.49 | 12.79 | 64.02 | 37.60 |
| | | Baseline | 1/32 | 35.8 | 8.43 | 13.11 | 65.45 | 29.88 |
| | | **BinaryDM** | 1/32 | 35.8 | **6.99** | **12.15** | **67.51** | **36.80** |
| | | Q-Diffusion | 2/8 | 69.8 | 62.01 | 33.56 | 16.48 | 14.12 |
| | | LSQ | 2/8 | 69.8 | 6.48 | 11.66 | 62.55 | 38.92 |
| | | Baseline | 1/8 | 35.8 | 9.37 | 12.10 | 64.36 | 30.76 |
| | | **BinaryDM** | 1/8 | 35.8 | **6.51** | **11.67** | **65.80** | **35.28** |
| | | Q-Diffusion | 4/4 | 134.9 | 427.46 | 277.22 | 0.00 | 0.00 |
| | | EfficientDM | 4/4 | 134.9 | 10.60 | - | - | - |
| | | LSQ | 2/4 | 69.8 | 12.95 | 12.79 | 55.97 | 34.30 |
| | | Baseline | 1/4 | 35.8 | 10.87 | 15.46 | 64.05 | 26.50 |
| | | TDQ | 1/4 | 35.8 | 11.28 | 12.80 | 55.14 | 27.32 |
| | | ReActNet | 1/4 | 35.8 | 10.23 | 13.02 | 61.43 | 29.68 |
| | | Q-DM | 1/4 | 35.8 | 9.99 | 11.96 | 57.62 | 29.30 |
| | | INSTA-BNN | 1/4 | 35.8 | 9.42 | 12.39 | 60.05 | 31.08 |
| | | BI-DiffSR | 1/4 | 35.8 | 8.58 | 11.81 | 62.61 | 30.86 |
| | | **BinaryDM** | 1/4 | 35.8 | **7.74** | **10.80** | **64.71** | **32.98** |
| LDM-8 | LSUN-Churches 256 × 256 | FP | 32/32 | 1125.2 | 4.82 | 17.66 | 75.18 | 46.80 |
| | | LSQ | 2/32 | 74.1 | 8.16 | 19.87 | 74.98 | 35.76 |
| | | Baseline | 1/32 | 38.1 | 9.91 | 17.94 | 74.89 | 26.88 |
| | | **BinaryDM** | 1/32 | 38.1 | **8.14** | **17.44** | **75.51** | **34.56** |
| | | Q-Diffusion | 2/8 | 74.1 | 201.23 | 238.70 | 2.39 | 8.60 |
| | | LSQ | 2/8 | 74.1 | 8.11 | 19.25 | 77.04 | 34.98 |
| | | Baseline | 1/8 | 38.1 | 10.94 | 16.95 | 74.30 | 25.66 |
| | | **BinaryDM** | 1/8 | 38.1 | **8.63** | **15.13** | **77.74** | **33.48** |
| | | EfficientDM | 4/4 | 144.2 | 14.34 | - | - | - |
| | | Q-Diffusion | 4/4 | 144.2 | 198.35 | 184.43 | 5.48 | 0.12 |
| | | LSQ | 2/4 | 74.1 | 10.00 | 19.08 | 74.93 | 25.80 |
| | | Baseline | 1/4 | 38.1 | 12.98 | 21.55 | 70.78 | 25.30 |
| | | **BinaryDM** | 1/4 | 38.1 | **9.91** | **18.04** | **73.72** | **29.96** |
| LDM-4 | FFHQ 256 × 256 | FP | 32/32 | 1045.4 | 6.64 | 14.16 | 76.88 | 50.82 |
| | | Q-Diffusion | 4/32 | 134.9 | 11.60 | 10.30 | - | - |
| | | Baseline | 1/32 | 35.8 | 10.49 | 11.56 | 72.64 | 39.62 |
| | | **BinaryDM** | 1/32 | 35.8 | **8.70** | **9.68** | **73.92** | **42.22** |
| | | Q-Diffusion | 8/8 | 265.0 | 10.87 | 10.01 | - | - |
| | | Q-Diffusion | 4/8 | 134.9 | 11.45 | 9.06 | - | - |
| | | Baseline | 1/8 | 35.8 | 10.79 | 10.77 | 73.20 | 41.70 |
| | | **BinaryDM** | 1/8 | 35.8 | **9.58** | **10.74** | **74.48** | **41.75** |
| | | Baseline | 1/4 | 35.8 | 15.07 | 12.48 | 74.34 | 35.12 |
| | | **BinaryDM** | 1/4 | 35.8 | **12.34** | **11.18** | **74.83** | **38.09** |

the binarized baseline by 9.46% in IS metrics on the CIFAR-10 and outperforms the LSQ under W2A4, where the latter involves several times of computation and storage.

Our LDM experiments encompass the evaluation of LDM-4 on LSUN-Bedrooms and FFHQ datasets, along with the assessment of LDM-8 on the LSUN-Churches dataset. The experiments utilized the DDIM sampler with 100 steps, and the detailed outcomes are presented in Table 2. We showcase results across various activation bit widths in the context of weight binarization, comparing them with the outcomes of some quantization methods at higher bit settings. The conventional binary baseline method exhibits subpar performance in the LDM context and experiences a further decline in the W1A4 experimental setup, particularly noticeable in the LSUN-Bedrooms dataset. In contrast, BinaryDM significantly enhances the generation quality, especially for LDM-4, exhibiting consistent performance across different activation bit settings. Notably, when compressing from W1A32 to W1A4 on the LSUN-Bedrooms dataset, the FID increased by a mere 0.75 for BinaryDM, showcasing its robustness. From the evaluation results of LDM-4 on FFHQ datasets, it can be observed that BinaryDM outperforms all other methods under various settings in terms of sFID, even surpassing W8A8 Q-Diffusion with a bit-width of W1A8. Moreover, BinaryDM demonstrates more significant

Table 3: Results on ImageNet $256 \times 256$ in conditional generation by DDIM with 20 steps.

| Sampler | Method | #Bits | IS↑ | FID↓ | sFID↓ | Prec.↑ |
|---|---|---|---|---|---|---|
| DDIM | FP | 32/32 | 235.84 | 12.96 | 25.99 | 92.63 |
| | Baseline | 1/32 | 197.85 | 11.50 | 23.44 | 84.83 |
| | **BinaryDM** | 1/32 | **215.55** | **10.86** | **21.10** | **88.43** |
| | Baseline | 1/8 | 203.90 | 11.35 | 25.49 | 85.78 |
| | **BinaryDM** | 1/8 | **211.43** | **11.23** | **24.12** | **88.09** |
| | Baseline | 1/4 | 187.70 | 11.51 | 20.77 | 84.13 |
| | **BinaryDM** | 1/4 | **208.42** | **10.78** | **20.40** | **87.61** |
| PLMS | FP | 32/32 | 247.38 | 13.54 | 18.85 | 94.22 |
| | Baseline | 1/32 | 211.69 | 11.23 | 21.32 | 86.16 |
| | **BinaryDM** | 1/32 | **226.86** | **11.00** | **19.01** | **91.17** |
| | Baseline | 1/8 | 205.58 | 12.78 | 21.57 | 84.07 |
| | **BinaryDM** | 1/8 | **225.18** | **11.33** | **19.18** | **90.35** |
| | Baseline | 1/4 | 193.11 | 11.08 | 23.21 | 81.40 |
| | **BinaryDM** | 1/4 | **218.06** | **10.36** | **18.85** | **88.74** |
| DPM-Solver | FP | 32/32 | 242.27 | 13.10 | 19.82 | 93.53 |
| | Baseline | 1/32 | 203.98 | 11.22 | 23.49 | 83.52 |
| | **BinaryDM** | 1/32 | **214.91** | **11.07** | **20.61** | **87.71** |
| | Baseline | 1/8 | 188.21 | 12.83 | 25.01 | 80.14 |
| | **BinaryDM** | 1/8 | **216.27** | **11.68** | **20.52** | **88.36** |
| | Baseline | 1/4 | 178.47 | 11.67 | 26.72 | 77.27 |
| | **BinaryDM** | 1/4 | **206.80** | **10.83** | **20.68** | **85.34** |

advantages at lower activation bit-widths, achieving accurate generation with an FID of 12.34 under 4-bit activation. BinaryDM even approaches the generation quality of the full-precision model, with specifically generated image examples provided in Appendix B.3.

**Conditional Generation.** For conditional generation, the performance of our BinaryDM is evaluated on the ImageNet dataset with a resolution of $256 \times 256$, focusing on LDM-4. We employ three distinct samplers to generate images: DDIM, PLMS, and DPM-Solver. The results in Table 3 underscore the remarkable effectiveness of our BinaryDM on DDIM, surpassing the baseline consistently across almost all evaluation metrics and even outperforming the full-precision diffusion model in several cases. The binarized DM baseline performs relatively stable in configurations W1A32 and W1A8 but significantly declines under W1A4, with the IS decreasing to 187.70 when using the DDIM sampler. In contrast, our BinaryDM maintains an IS of 208.42 in W1A4. Specifically, when utilizing the DPM-Solver sampler, the IS plummets to 178.47, and the sFID experiences a sharp increase to 26.72. In stark contrast, our binarized DM maintains consistently high performance, achieving a 206.80 IS and a 20.68 FID and outperforming the baseline in most scenarios.

## 4.2 ABLATION STUDY

We perform comprehensive ablation studies for LDM-4 on the LSUN-Bedrooms dataset to evaluate the effectiveness of our proposed EBB and LRM, and the results are presented in Table 4.

The performance has shown significant recovery when first applying our EBB to binarized diffusion models, with the FID decreasing from 8.43 to 7.39. This confirms that the degradation in parameter representational capacity due to binarization is a primary performance bottleneck in the binarized DM baseline. Solving this representation degradation is a prerequisite for improving model performance. From a structural perspective, EBB provides binarized diffusion models with an initial state with a higher information capacity, alleviating the degradation of representational ability in the early stages and guiding QAT toward a more easily optimizable direction.

With the application of LRM on this basis, the generative capability of the resulting binarized diffusion models is further enhanced, with the FID decreasing to 6.99. This indicates that the low-rank mimicking scheme, designed from a feature-matching perspective, effectively utilizes the representational information of the full-precision model, achieving supervision and alignment of intermediate features and better guiding the optimization of the binarized diffusion models.

Further substantiating this view, the detailed ablation experiments in Appendix B.2 delve into an in-depth discussion of the specifics concerning EBB and LRM. Combining these two techniques in

Table 4: Ablation results on LSUN-Bedrooms $256 \times 256$.

| Method | #Bits | FID↓ | sFID↓ | Prec.↑ | Recall↑ |
|--------|-------|------|-------|--------|---------|
| FP | 32/32 | 3.09 | 7.08 | 65.82 | 45.36 |
| Vanilla | 1/32 | 8.43 | 13.11 | 65.45 | 29.88 |
| +EBB | 1/32 | 7.39 | 12.34 | 65.98 | 35.84 |
| +LRM | 1/32 | **6.99** | **12.15** | **67.51** | **36.80** |

BinaryDM can significantly enhance performance, emphasizing that a better optimization process can improve the quality of generation when ensuring accurate representation.

### 4.3 EFFICIENCY ANALYSIS

For inference, we demonstrate the size and OPs of BinaryDM under different activation bit-widths. The results in Table 5 indicate that our DM can achieve up to 29.2× space savings while obtaining up to 15.2× acceleration during inference, fully harnessing the advantages of binary computation. BinaryDM achieves optimal inference efficiency while surpassing the performance of advanced methods with higher bit widths. The W1A1 BinaryDM achieves a lower FID compared to the W4A4 EfficientDM, while its model size and OPs are only 26.5% and 25.9% of the latter, respectively.

For training, while the training process for our binarized DM typically incurs higher overhead compared to post-training quantization methods, practical observations reveal that our approach offers productivity advantages across various models and datasets. As shown in Table 6, despite having a training time shorter than the calibration time required by Q-Diffusion, our method attains significantly superior generation quality, particularly at lower bits.

Table 5: Inference efficiency of our proposed BinaryDM of LDM-4 on LSUN-Bedrooms $256 \times 256$.

| Model | Method | #Bits | Size$_{(MB)}$ | OPs$_{(\times 10^9)}$ | FID↓ |
|-------|--------|-------|---------------|------------------------|------|
| LDM-4 | Full-Precision | 4/4 | 1045.4 | 96.0 | 3.09 |
| | Q-Diffusion | 4/4 | 134.9 | 24.3 | 427.46 |
| | EfficientDM | 4/4 | 134.9 | 24.3 | 10.60 |
| | LSQ | 2/4 | 69.8 | 12.3 | 12.95 |
| | **BinaryDM** | 1/4 | **35.8** | **6.3** | **7.74** |

Table 6: Training time-cost of BinaryDM compared to the advanced PTQ method.

| Dataset | Method | #Bits | Size$_{(MB)}$ | Time$_{(h)}$ | FID↓ |
|---------|--------|-------|---------------|--------------|------|
| LSUN-Bedrooms | Q-Diffusion | 4/4 | 134.9 | 13.7 | 427.46 |
| | **BinaryDM** | 1/4 | **35.8** | **11.3** | **13.93** |
| LSUN-Churches | Q-Diffusion | 4/4 | 144.2 | 10.9 | 198.35 |
| | **BinaryDM** | 1/4 | **38.1** | **9.0** | **15.11** |

## 5 CONCLUSION

In this paper, we propose BinaryDM, a novel accurate quantization-aware training approach to push the weights of diffusion models towards the limit of binary. Firstly, we present an Evolvable-Basis Binarizer (EBB) to enable the QAT of binarized DMs to start from a more favorable initial state, leading to a smoother optimization process and better final results. Secondly, a Low-rank Representation Mimicking (LRM) is applied to enhance the binarization-aware optimization of the DM, alleviating the optimization direction ambiguity caused by fine-grained alignment. Comprehensive experiments demonstrate that BinaryDM achieves significant accuracy and efficiency gains compared to SOTA quantization methods of DMs under ultra-low bit-widths. As the first binarization method for diffusion models, W1A4 BinaryDM achieves impressive 15.2× OPs and 29.2× storage savings, showcasing substantial advantages and potential for deploying DMs on edge.

**Acknowledgement** This work was supported by the Beijing Municipal Science and Technology Project (No. Z231100010323002), the National Natural Science Foundation of China (Nos. 62306025, 92367204), CCF-Baidu Open Fund, Beijing Natural Science Foundation (QY24138), the Swiss National Science Foundation (SNSF) project 200021E_219943 Neuromorphic Attention Models for Event Data (NAMED), and Baidu Scholarship.

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

## A    DETAILS OF BINARYDM

**Diffusion Models** have showcased remarkable performance across a diverse array of tasks (Ho et al., 2020; Song & Ermon, 2019; Song et al., 2020b; Niu et al., 2020; Mittal et al., 2021; Popov et al., 2021; Jeong et al., 2021). These tasks involve a forward Markov chain process, wherein generated noisy samples are incrementally added through Gaussian noise. Subsequently, a reverse denoising process refines these samples, producing high-fidelity images. However, the diffusion model's slow generation process poses a significant challenge to widespread implementation. To address this issue, substantial research has concentrated on reducing the time steps required for diffusion model generation. Techniques such as trajectory search can be formulated as dynamic programming problems (Watson et al., 2021), and grid search has demonstrated the ability to discover effective trajectories within a mere six-time steps (Chen et al., 2020). Moreover, the introduction of non-Markov diffusion processes has been instrumental in expediting sampling during the reverse process (Song & Ermon, 2019; Song et al., 2020b), with the application of numerical methods to solve associated equations resulting in a notable reduction in the number of iterations to just a few dozen. Efforts to address these challenges have led to exploring faster step size schedules for VP diffusions, demonstrating the ability to maintain relatively good quality and diversity metrics (San-Roman et al., 2021). Additionally, analytical approximations have been derived to simplify the generation process (Bao et al., 2022). These developments mark strides towards enhancing the efficiency and practicality of diffusion models in various applications. Despite these advancements, the denoising models still involve a considerable number of parameters, demanding substantial computation and memory resources for each denoising step. This computational expense hinders the practical implementation of the inference process on standard hardware.

**Quantization and Binarization** are popular compression approaches (Nagel et al., 2020; **?**; **?**; Lin et al., 2021; Huang et al., 2024b; Qin et al., 2023a), which quantize the full-precision parameters of the neural network to lower bit-width (*e.g.*, 1-8 bit). By converting floating-point weight and activation into quantized ones, the model size of the neural network can be decreased, and the computational complexity can also be reduced, leading to significant inference speedup, memory usage savings, and lower energy consumption. Model quantization methods for diffusion models are generally divided into two categories based on their pipeline and resource access during training or fine-tuning: post-training quantization (Shang et al., 2023; Li et al., 2023a) and quantization-aware training (Li et al., 2023b; He et al., 2023; Feng et al., 2024a). As a training-free method, post-training quantization is considered a more practical solution to obtain quantized models at low cost by searching for the best scaling factor candidates and optimizing the calibration strategy. However, the diffusion models quantized by post-training methods dramatically degrade generation quality (Shang et al., 2023; Li et al., 2023a). Thus, quantization-aware training emerges for pushing quantized neural networks to higher accuracy (So et al., 2024; Li et al., 2023b; He et al., 2023). Benefiting from the training/fine-tuning process with sufficient data and training resources, the low-bit diffusion model obtained by quantization-aware training methods usually achieves higher accuracy than post-training ones. However, binarization for the weight of diffusion models is still far from available since it suffers serious accuracy degeneration challenges in existing methods.

## B    EXPERIMENTS AND VISUALIZATION

### B.1    EXPERIMENT SETTINGS

**Experimental Hardware.** All our experiments were conducted on a server with Intel Xeon Gold 6336Y 2.40@GHz CPU and NVIDIA A100 40GB GPU.

**Models and Dataset.** We perform comprehensive experiments encompassing unconditional image generation and conditional image generation tasks on two diffusion models: pixel-space diffusion model DDIM and latent-space diffusion model LDM. Specifically, we conduct experiments on DDIM using the CIFAR-10 dataset with a resolution of $32 \times 32$. For LDM, our investigations spanned multiple datasets, including LSUN-Bedrooms, LSUN-Churches, and FFHQ, all with a resolution of $256 \times 256$. Furthermore, we employ LDM for conditional image generation on the ImageNet dataset with a resolution of $256 \times 256$. This diverse set of experiments, conducted on different models, datasets, and tasks, allows us to comprehensively validate our proposed method's effectiveness.

**Proposed Quantization Baselines.** We use per-channel quantizers for weights and per-layer quantizers, as is a common practice. To the best of our knowledge, the weights of the diffusion model have not yet been binarized, and we found in our initial attempts that the basic BNN without scaling factors would collapse directly at the beginning of training. Hence, we utilize the fundamental binary quantizer, as outlined in Section 3.1, as the baseline for weight quantization. LSQ serves as the foundational method for activations quantization. Under the uniform premise that the weights are binarized, we use a variety of quantization bit-widths for activations to cover as many realistic situations as possible. It's crucial to emphasize that we only quantize the diffusion model without quantizing the VAE part of the LDM. Additionally, we quantized the layers closest to the input and output to 8-bit, adhering to a common practice in this context.

**Compared Advanced Strategies.** In addition to the Baseline strategy we constructed, we also compared BinaryDM against advanced general binarization strategies and quantization strategies for diffusion models. The advanced general quantization strategies include the low-bit approach LSQ (Esser et al., 2019), as well as binarization strategies such as ReActNet (Liu et al., 2020) and INSTA-BNN (Lee et al., 2023). Quantization strategies for diffusion models include the PTQ strategy Q-Diffusion (Li et al., 2023a) and QAT strategies such as EfficientDM (He et al., 2023), Q-DM (Li et al., 2024), TDQ (So et al., 2024), and BI-DiffSR (Chen et al., 2024b). These comparisons with advanced methods highlight the effectiveness of BinaryDM.

**Pipeline and hyperparameters.** Our quantization-aware training (QAT) is based on the pre-trained diffusion model, and the quantizer parameters and latent weights are trained simultaneously. The overall training process is relatively consistent with the original training process of DDIM or LDM. Relative to the training hyperparameters of the full precision model, we adjust the learning rate, reducing it to one-tenth to one-hundredth of the corresponding rate in the original full precision training script, especially on certain datasets, such as CIFAR-10 and ImageNet. For DDIM training, we set the batch size to 64, while for LDM training, the batch size is configured as 4. Typically, models are trained for around 200K iterations.

**Evaluation.** To assess the generation quality of the diffusion model, we utilize several evaluation metrics, including Inception Score (IS), Fréchet Inception Distance (FID), Sliding Fréchet Inception Distance (sFID), and Precision-and-Recall. We randomly generate 50,000 samples from the model in each evaluation and compute the metrics using reference batches. The reference batches used to evaluate FID and sFID contain all the corresponding datasets, while only 10,000 images were extracted when Precision and Recall were calculated. These metrics are all evaluated using ADM's TensorFlow evaluation suite.

**Efficiency.** We utilize Time and OPs as metrics for evaluating training efficiency and theoretical inference efficiency, respectively. For OPs, taking the convolutional unit as an example, the BOPs definition for binary convolution operations is as follows (Yang et al., 2024; Wang et al., 2021):

$$BOPs \approx whmnk^2 b_a b_w. \tag{16}$$

It is composed of $b_w$ bits for weights, $b_a$ bits for activation, $n$ input channels, $m$ output channels, a $k \times k$ convolutional kernel, and output dimensions of width $w$ and height $h$ for each channel. As there might also be full-precision modules in the model, the total OPs of the model are summed up according to the following method (Bethge et al., 2020):

$$OPs = \left(\frac{1}{64} BOPs + FLOPs\right). \tag{17}$$

### B.2 ADDITIONAL RESULTS

**Further Ablation Study on W1A4.** The ablation experiments in our main text were conducted on W1A32, as the highlight of BinaryDM lies in achieving weight binarization for DM, with the activation quantization method always using a naive scheme without any additional complex techniques. Here, we also supplement the ablation results on the more efficient W1A4 model. As shown in the Table 7, when EBB was added alone, the generative performance of the binary DM improved significantly, with the FID decreasing from 10.87 to 8.53. After adding LRM, the FID further decreased to 7.74, clearly illustrating the effectiveness of their synergistic effect.

**Effects of EBB.** We conducted comprehensive experiments on various aspects of EBB's specific details to validate its effectiveness further.

Table 7: Ablation results on LSUN-Bedrooms $256 \times 256$.

| Method | #Bits | FID↓ | sFID↓ | Prec.↑ | Recall↑ |
|--------|-------|------|-------|--------|---------|
| FP | 32/32 | 3.09 | 7.08 | 65.82 | 45.36 |
| Vanilla | 1/4 | 10.87 | 15.46 | 64.05 | 26.50 |
| +EBB | 1/4 | 8.53 | 11.99 | 62.94 | 30.78 |
| +LRM | 1/4 | **7.74** | **10.80** | **64.71** | **32.98** |

As a supplement to the ablation study on the final generation performance (Table 4) in the main text, we present in Table 8 the changes in training loss ($\mathcal{L}_{\text{simple}}$) at different iterations. The results indicate that EBB consistently achieves lower training loss, demonstrating its benefits for convergence.

Table 8: Training loss ($\mathcal{L}_{\text{simple}}$) at different iterations on LSUN-Bedrooms, comparing the baseline and the addition of EBB.

| Method | #Bits | Iterations | | | | |
|--------|-------|------|------|------|------|------|
| | | 1e1 | 1e2 | 1e3 | 1e4 | 1e5 |
| Baseline | 1/32 | 0.388 | 0.303 | 0.277 | 0.227 | 0.158 |
| **+EBB** | 1/32 | **0.352** | **0.264** | **0.242** | **0.206** | **0.151** |

The results in Table 9 and Table 10 demonstrate that applying EBB significantly improves the generative quality of binarized diffusion models, highlighting the effectiveness of EBB. Furthermore, not applying EBB to the *Central Parts* yields better optimization results. The results in Table 10 demonstrate that applying EBB significantly improves the generative quality of binarized diffusion models, highlighting the effectiveness of EBB. Furthermore, not applying EBB to the *Central Parts* yields better optimization results. This suggests that applying EBB only to the key parts reduces the number of parameter updates when transitioning to the second stage, thus leading to a more stable optimization process for binarized diffusion models. Specifically, applying EBB to regions with high parameter counts but lower sensitivity to binarization can lead to suboptimal optimization stability, resulting in worse performance compared to applying EBB selectively. Additionally, while Head and Tail Parts (12) achieves lower training loss in the first 1000 iterations compared to Head and Tail Parts (6), its weaker transition to full weight binarization results in higher loss at 100K iterations. This suggests that applying EBB only to the key parts reduces the number of parameter updates when transitioning to the second stage, thus leading to a more stable optimization process for binarized diffusion models.

Table 9: The impact of EBB application scopes on LSUN-Bedrooms (1/2), where *Head and Tail Parts* refers to how many of the first and last Timestep Embed Blocks and *Central Parts* refers to the middle Blocks.

| Head and Tail Parts | Central Parts | #Bits | Iterations | | | | |
|---------------------|---------------|-------|------|------|------|------|------|
| | | | 1e1 | 1e2 | 1e3 | 1e4 | 1e5 |
| 0 | 0 | 1/32 | 0.335 | 0.291 | 0.268 | 0.202 | 0.141 |
| 3 | 0 | 1/32 | 0.335 | 0.263 | 0.230 | 0.184 | 0.138 |
| **6** | **0** | 1/32 | 0.332 | 0.238 | 0.199 | **0.178** | **0.130** |
| 0 | 0 | 1/32 | 0.331 | 0.223 | 0.201 | 0.183 | 0.133 |
| 12 | 1 | 1/32 | 0.331 | 0.225 | 0.197 | 0.188 | 0.136 |

We conducted extensive experiments to verify the impact of the regularization loss coefficient $\mu$ on training, as shown in Table 11. Here, $\mu = 0$ indicates that no regularization penalty is applied in the first stage, and the second learnable scalar $\sigma_{\text{II}}$ is directly removed at the beginning of the second stage. The results demonstrate that the transition process using the regularization strategy leads to better optimization outcomes for the binarized DM. Furthermore, EBB shows good robustness to $\mu$, with a moderately larger $\mu$ yielding better final generative performance.

Table 10: The impact of EBB application scopes on LSUN-Bedrooms (2/2).

| Head and Tail Parts | Central Parts | #Bits | FID↓ | sFID↓ | Precision↑ | Recall↑ |
|---|---|---|---|---|---|---|
| 0 | 0 | 1/32 | 8.02 | 12.81 | 64.83 | 33.12 |
| 3 | 0 | 1/32 | 7.20 | 12.27 | 65.62 | 34.98 |
| **6** | **0** | 1/32 | **6.99** | **12.15** | **67.51** | **36.80** |
| 0 | 0 | 1/32 | 7.10 | 12.22 | 65.41 | 36.42 |
| 12 | 1 | 1/32 | 7.10 | 12.29 | 66.41 | 34.54 |

Table 11: The impact of the regularization loss coefficient $\mu$ on LSUN-Bedrooms $256 \times 256$.

| $\mu$ | #Bits | FID↓ | sFID↓ | Prec.↑ | Recall↑ |
|---|---|---|---|---|---|
| 0 | 1/32 | 8.01 | 13.16 | 64.34 | 30.06 |
| **9e-2** | 1/32 | **6.99** | **12.15** | **67.51** | **36.80** |
| 9e-3 | 1/32 | 7.26 | 12.26 | 65.10 | 34.44 |
| 9e-4 | 1/32 | 7.18 | 11.83 | 66.96 | 34.54 |

We conducted experiments on the timing of EBB's transition to the second stage. In Table 12, an iteration of 0 indicates that EBB is not applied. The results demonstrate the effectiveness of EBB and the transition strategy with regularization penalties, with a slightly longer regularization phase yielding marginally better final generative outcomes for binarized DMs.

Table 12: The impact of the iteration at which EBB transitions to the second stage on LSUN-Bedrooms $256 \times 256$.

| Iterations | #Bits | FID↓ | sFID↓ | Prec.↑ | Recall↑ |
|---|---|---|---|---|---|
| 0 | 1/32 | 8.22 | 13.02 | 61.45 | 32.88 |
| 10000 | 1/32 | 7.08 | 12.30 | 64.99 | 36.18 |
| **100000** | 1/32 | **6.99** | **12.15** | **67.51** | **36.80** |

**Further Discussion of EBB.** From the perspective of the final optimization outcome, in Eq.5, removing the sign function from the second term—i.e., replacing $\sigma_{\mathrm{II}} \operatorname{sign} (\boldsymbol{w} - \sigma_1 \operatorname{sign} (\boldsymbol{w}))$ with $\sigma_{\mathrm{II}} (\boldsymbol{w} - \sigma_1 \operatorname{sign} (\boldsymbol{w}))$—can also achieve the same final evolutionary state. However, although this modification may provide stronger fitting ability in the early stages, it is not beneficial to achieve a fully binarized final model through optimization. This is because it would lead to a significant imbalance between the representations of the initial and final stages, causing the second term corresponding to $\sigma_{\mathrm{II}}$ to dominate due to its excessive information extraction capability, which in turn hinders the subsequent optimization of $\sigma_1 \operatorname{sign} (\boldsymbol{w})$.

The additional experimental results we present in the Table 13 show that this idea did not achieve the optimal generation performance. At the same time, we observed the value of $\sigma_{\mathrm{II}}$ at the end of the first phase of EBB training. We found that, unlike the original approach where regularization would force it to converge to nearly zero, $\sigma_{\mathrm{II}}$ still had a relatively large value, which indicates that this idea indeed hindered the natural evolution from multiple bases to a single base, and reduced the effectiveness of EBB as a method designed to enhance learning ability.

Table 13: The impact of applying the sign function to the second term in EBB on LSUN-Bedrooms $256 \times 256$.

| Method | #Bits | FID↓ | sFID↓ | Prec.↑ | Recall↑ |
|---|---|---|---|---|---|
| Baseline | 1/4 | 10.87 | 15.46 | 64.05 | 26.50 |
| no sign in second term | 1/4 | 8.44 | 13.00 | 62.68 | 30.12 |
| **BinaryDM** | 1/4 | **7.74** | **10.80** | **64.71** | **32.98** |

**Effects of LRM.** As a supplement to the ablation study on the final generation performance (Table 4) in the main text, we present in Table 14 the changes in training loss ($\mathcal{L}_{\text{simple}}$) at different iterations. The results indicate that LRM consistently achieves lower training loss, demonstrating its benefits for convergence.

Table 14: Training loss ($\mathcal{L}_{\text{simple}}$) at different iterations on LSUN-Bedrooms, comparing the no distillation, MSE and the addition of LRM.

| Method | #Bits | Iterations | | | | |
|---|---|---|---|---|---|---|
| | | 1e1 | 1e2 | 1e3 | 1e4 | 1e5 |
| Baseline | 1/32 | 0.388 | 0.303 | 0.277 | 0.227 | 0.158 |
| $\mathcal{L}_{\text{MSE}}$ | 1/32 | 0.388 | 0.303 | 0.277 | 0.227 | 0.158 |
| $\mathcal{L}_{\textbf{LRM}}$ | 1/32 | **0.352** | **0.264** | **0.242** | **0.206** | **0.151** |

We evaluate the performance of our binarized diffusion model under various values of $K$ (reduction times of dimension) when incorporating LRM. Additionally, we compare these results with the outcomes of applying MSE distillation directly to the output features of blocks without dimensionality reduction. The experiments reveal the model's generation capability improves effectively when an appropriate degree of dimension reduction is employed, as illustrated in Table 15.

Table 15: In the application of LRM, the impact of different reduction times of dimension on the experimental results on LSUN-Bedrooms $256 \times 256$.

| $\mathcal{L}_{\text{distil}}$ | $K$ | #Bits | FID↓ | sFID↓ | Prec.↑ | Recall↑ |
|---|---|---|---|---|---|---|
| - | - | 1/32 | 7.39 | 12.34 | 65.98 | 35.84 |
| $\mathcal{L}_{\text{MSE}}$ | - | 1/32 | 7.36 | 12.76 | 62.05 | 33.64 |
| | 2 | 1/32 | 7.21 | 12.22 | 65.86 | 36.00 |
| $\mathcal{L}_{\text{LRM}}$ | 4 | 1/32 | 6.99 | 12.15 | **67.51** | **36.80** |
| | 8 | 1/32 | **6.95** | **12.02** | 64.20 | 35.44 |

As an additional clarification on stability, we also conducted experiments where the dimensionality reduction matrix $E_i^{\lceil \frac{c}{K} \rceil}$ is updated every 100 iterations. As shown in the Table 16, while using LRM consistently yields improvements (with FID decreasing from 7.39 to 7.11/6.99), the approach of initializing the matrix once and retaining it throughout results in the highest accuracy. This further confirms our analysis that fixing the dimensionality reduction matrix and not updating it is more beneficial for stable optimization.

Table 16: Results of different update frequency of LRM on LSUN-Bedrooms.

| Update Frequency (/iter) | #Bits | FID↓ | sFID↓ |
|---|---|---|---|
| 0 (w/o LRM) | 1/32 | 7.39 | 12.34 |
| 100 | 1/32 | 7.11 | 12.23 |
| $\infty$ (**BianryDM**) | 1/32 | **6.99** | **12.15** |

**Further Efficiency Analysis.** We pointed out in the main text that certain high-order-based structures are computationally unfriendly. In fact, The models produced by our method save 1.96x in parameters (Size) and 2.00x in computational operations (OPs) during inference, and we have also provided hardware implementations. Specifically, methods based on higher-order residual bases require more sets of binarized weights and corresponding scaling factors during inference compared to Baseline or BinaryDM (Eq.10):

$$\boldsymbol{w}^{\text{bi}} = \sigma_{\text{I}}\left(\boldsymbol{w}_I^{bi}\right) + \sigma_{\text{II}}\left(\boldsymbol{w}_{II}^{bi}\right). \tag{18}$$

This at least doubles the parameter count and OPs. Additionally, although multiple sets of bases in higher-order methods are expected to be processed in parallel during inference, we found in our research that, to date, there has not been any implementation of this, making them computationally less efficient.

For actual hardware, we implemented convolution and linear layers unit by unit to estimate the overall model, utilizing the general deployment library Larq[1] on a Qualcomm Snapdragon 855 Plus to test the actual runtime efficiency of the aforementioned single convolution. Since the current deployment libraries do not support direct computation for W1A4, we used a combined approach to achieve it via W1A1. Specifically, for the W1A4 operator, since there is no existing 4-bit activation implementation, we decompose the activation as follows:

$$k \cdot a^{4bit} = 4k \cdot b_{a_1}^{1bit} + 2k \cdot b_{a_2}^{1bit} + k \cdot b_{a_3}^{1bit} + \frac{1}{2}k \cdot b_{a_4}^{1bit} - \frac{1}{2}k, \qquad (19)$$

where

- $k$ is the scaling factor of fp32 activation,
- $a^{4bit} \in \{-8, -7, -6, -5, -4, -3, -2, -1, 0, 1, 2, 3, 4, 5, 6, 7\}$,
- $b_{a_i}^{1bit} \in \{-1, 1\}, i \in \{1, 2, 3, 4\}$.

As a result, the computation of 1-bit weights with int4 can be straightforwardly decomposed into the computation of 1-bit weights with 4 1-bit activations and one bias term ($\frac{1}{2}k$), based on the W1A1 operator provided by Larq, with the addition of limited arithmetic operations. The runtime results for a single inference are summarized in the Table 17. Due to limitations of the deployment library and hardware, Baseline/BinaryDM achieved a 4.62x speedup, while High-Order only achieved an 3.11x speedup. With further hardware support for binary operations, BinaryDM is expected to achieve performance closer to the theoretical OPs calculations (15.2x), further widening the gap between its implementation and that of high-order methods.

Table 17: The actual runtime efficiency of a single convolution.

| Method | #Bits | Size(MB) | Theoretical OPs($\times 10^9$) | Runtime($\mu s$/convolution) |
|---|---|---|---|---|
| FP | 32/32 | 1045.4 | 96.0 | 176371.0 |
| High-Order | 1/4 | 70.2 | 12.6 | 56657.5 |
| **BinaryDM** | 1/4 | **35.8** | **6.3** | **38174.2** |

## B.3 VISUALIZATION RESULTS

**Visualization of the impact of LRM.** As a complement to Figure 3, we present the distance in output features between binary DM and full-precision DM on more blocks under different distillation losses. As shown in Figure 5, our proposed PCA-based distillation strategy consistently possesses the optimal guiding constraint capability.

**Additional Random Samples.** We showcase random generation results on various datasets, with unconditional generation on LSUN-Bedrooms, LSUN-Churches, and FFHQ datasets, and conditional generation on ImageNet. Overall, BinaryDM exhibits the best generation performance across datasets and maintains relatively stable performance as the activation bit-width decreases from 32 to 4 bits. In contrast, the Baseline lacks detailed textures and experiences significant performance degradation as the activation bit-width decreases.

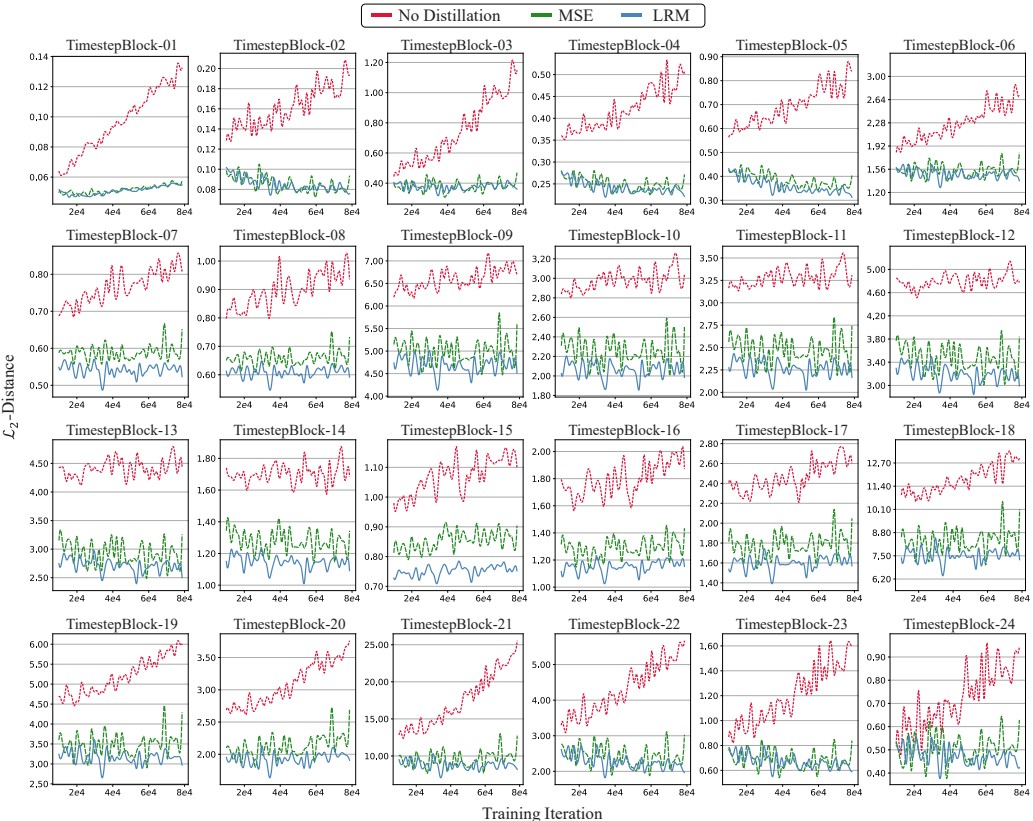

Figure 5: A comprehensive record of the impact of different distillation loss functions on the output features of each block in both full-precision DM and binarized DM, measured using the $\mathcal{L}_2$ distance.

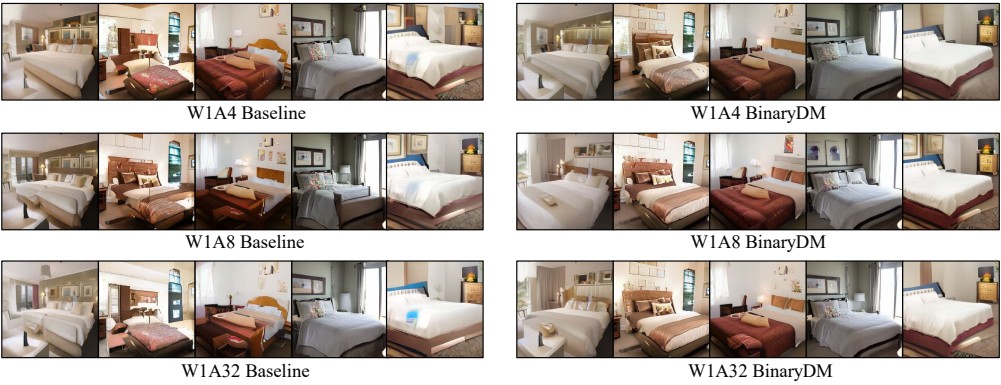

Figure 6: Samples generated by BinaryDM and Baseline on LSUN-Bedrooms 256 x 256

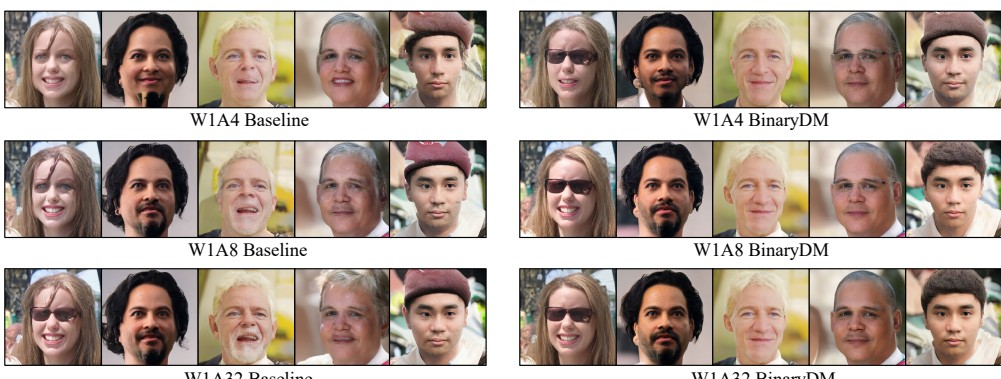

Figure 7: Samples generated by BinaryDM and Baseline on FFHQ 256 x 256

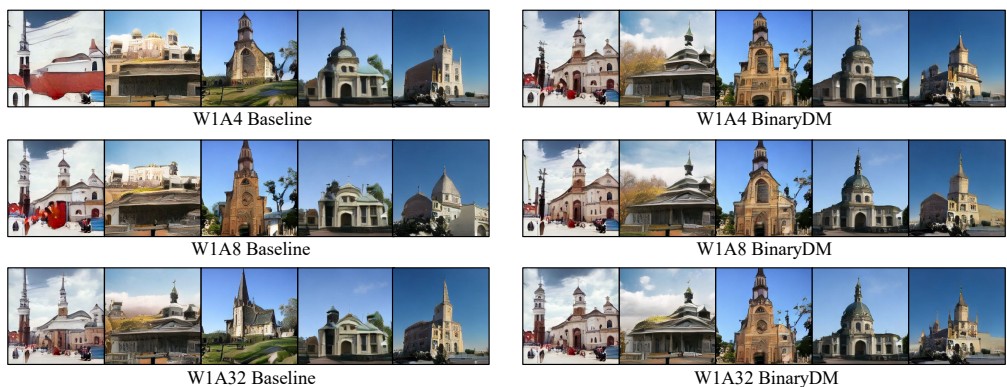

Figure 8: Samples generated by BinaryDM and Baseline on LSUN-Churches 256 x 256

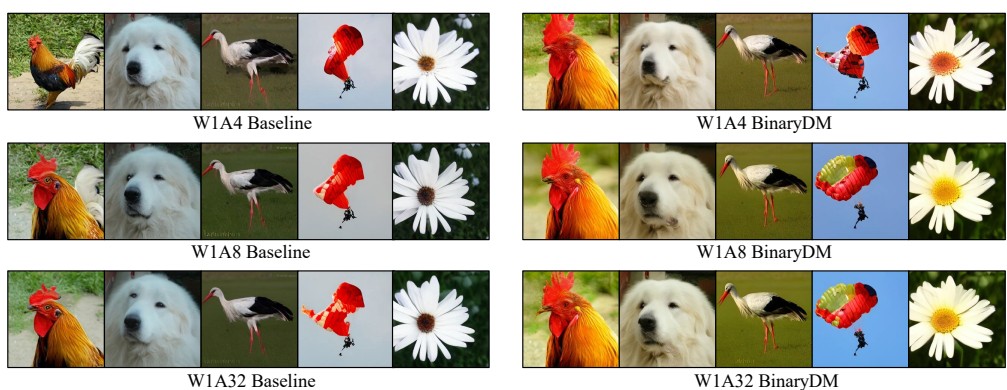

Figure 9: Samples generated by BinaryDM and Baseline on ImageNet 256 x 256

