# OpenReview forum: "BinaryDM: Accurate Weight Binarization for Efficient Diffusion Models"
_ICLR.cc/2025/Conference — ICLR 2025 Poster_

### Official Review · Reviewer_riMU · 2024-11-01

**Soundness:** 3
**Presentation:** 2
**Contribution:** 2
**Rating:** 6
**Confidence:** 4

**Summary:**

In this work, the authors propose a quantization-aware training method that targets ultra-low bit-width representations, down to 1 bit resolution, oriented to diffusion models. More specifically, the proposed method consists of two parts, one focused on the feature representation of the applied diffusion architecture and the other on the optimization process. From the representation perspective, the authors propose a novel mechanism that gradually reduces the bases of binarized weights from higher to single-order base, applying a regularization term that according to the authors allows to smoothen the binarization process offering rich representation during the initial stage of training. Furthermore, the authors introduce a binarization-aware optimization method that projects the binarized and full-precision representations to lower ranks, applying Principal Component Analysis, forcing the low-rank binary representation to mimic the full precision one. This, according to the authors, enables the optimization of binarized diffusion models to focus on the principal direction and mitigate the ambiguity of the direction caused by the representation complexity of generation. The proposed method is applied in supplement with the vanilla binarization, with the proposed method targeting the most sensitive modules of the architecture, meaning on the outer layers, with the vanilla binarization applied on the intermediate layers. The paper provides extensive experimental evaluation on various datasets, including ablation studies, showing the effectiveness of the proposed method in contrast to the applied baselines, with the authors claiming that the applied methodology can significantly reduce the required Point Operations during the inference.

**Strengths:**

The proposed method results in significantly lower sized models with improved performance in reference to the baselines. This can potentially lead to significantly lower point operations and memory size during hardware deployment.

The proposed Evolvable-Basis Binarized results in richer representation during the initial stage of training as reported in Figure 2, showing significantly higher variance of parameters during the first iterations. As far I can say, the proposed method in fact smoothen the optimization process.

In general, the proposed method achieves sufficient performance in the challenging task of binarization.

**Weaknesses:**

The authors mention that the existing works have enhanced the binarized parameters via higher-order residual bases, but they introduce additional hardware complexity. However, the paper does not provide any evidence that the proposed method can be actually deployed in hardware accelerators without additional hardware, while the efficiency analysis reported is based on an estimation of OPs. The authors should note on the efficiency analysis section of the paper that is based on estimation and it would be beneficial to further discuss how their proposed method benefits the hardware inference in contrast to the existing methodologies. Can the authors elaborate on that, what overhead is introduced in existing methods in contrast to the proposed one?

Without theoretical nor empirical analysis regarding the location selection, I am concerned about how the authors conclude on applying the proposed approach on the first and last six layers of the models. Does the proposed method can be applied out-of-the-box on other architectures without applying exhaustive search on which layers should be applied? This makes me even more concerned regarding the generality of the proposed method is the fact that the proposed method is computationally intensive since it applies quantization-aware training that requires the computation of the covariance matrix of the binarized and full-precision representation. This introduces significant limitations on searching the layers where the application of the proposed method could be beneficial, hindering its generalization ability.

The notation used in some cases is weak. For example, it is not clear whether Equation 6 is applied channel-wise. Additionally, in the case of the trainable parameter $\sigma_{1}$ the domain is missing, raising the question whether the sigma can change the polarity of the binarized parameter or not. Additionally, it is not clear in the text if the $\sigma_{1}$ parameter is discarded after the training or if it is used also during the inference. Furthermore, using the same notation to denote different parameters as in the cases of Equations 9 and 2 can be misleading. Finally, $\epsilon_t$ is not properly defined in the text.

Although the most interesting part of the proposed method is the effective bit reduction down to 1W4A, the ablation study, which evaluates the different parts of the proposed method, is conducted only for 1W32A. I strongly recommend the authors to also include an ablation study for 1W4A, as it is the most impressive achievement of the proposed method.

**Questions:**

The authors mention that the existing works have enhanced the binarized parameters via higher-order residual bases, but they introduce additional hardware complexity. However, the paper does not provide any evidence that the proposed method can be actually deployed in hardware accelerators without additional hardware, while the efficiency analysis reported is based on an estimation of OPs. The authors should note on the efficiency analysis section of the paper that is based on estimation and it would be beneficial to further discuss how their proposed method benefits the hardware inference in contrast to the existing methodologies. Can the authors elaborate on that, what overhead is introduced in existing methods in contrast to the proposed one?

Without theoretical nor empirical analysis regarding the location selection, I am concerned about how the authors conclude on applying the proposed approach on the first and last six layers of the models. Does the proposed method can be applied out-of-the-box on other architectures without applying exhaustive search on which layers should be applied? This makes me even more concerned regarding the generality of the proposed method is the fact that the proposed method is computationally intensive since it applies quantization-aware training that requires the computation of the covariance matrix of the binarized and full-precision representation. This introduces significant limitations on searching the layers where the application of the proposed method could be beneficial, hindering its generalization ability.

The notation used in some cases is weak. For example, it is not clear whether Equation 6 is applied channel-wise. Additionally, in the case of the trainable parameter $\sigma_{1}$ the domain is missing, raising the question whether the sigma can change the polarity of the binarized parameter or not. Additionally, it is not clear in the text if the $\sigma_{1}$ parameter is discarded after the training or if it is used also during the inference. Furthermore, using the same notation to denote different parameters as in the cases of Equations 9 and 2 can be misleading. Finally, $\epsilon_t$ is not properly defined in the text.

Although the most interesting part of the proposed method is the effective bit reduction down to 1W4A, the ablation study, which evaluates the different parts of the proposed method, is conducted only for 1W32A. I strongly recommend the authors to also include an ablation study for 1W4A, as it is the most impressive achievement of the proposed method.

---

> ### Author Response · Authors · 2024-11-21
> **Response to Reviewer riMU**
>
> Thank you for your careful review of our work. We respond to the concerns as below:
>
> > **Q1.** The authors mention that the existing works have enhanced the binarized parameters via higher-order residual bases, but they introduce additional hardware complexity. However, the paper does not provide any evidence that the proposed method can be actually deployed in hardware accelerators without additional hardware, while the efficiency analysis reported is based on an estimation of OPs. The authors should note on the efficiency analysis section of the paper that is based on estimation and it would be beneficial to further discuss how their proposed method benefits the hardware inference in contrast to the existing methodologies. Can the authors elaborate on that, what overhead is introduced in existing methods in contrast to the proposed one?
>
> **A1.** Thank you for your suggestion. We would like to clarify that our method is more efficient than higher-order methods, both theoretically and in actual deployment. The models produced by our method save 1.96x in parameters (Size) and 2.00x in computational operations (OPs) during inference, and we have also provided hardware implementations based on your suggestion.
>
> Specifically, methods based on higher-order residual bases require more sets of binarized weights and corresponding scaling factors during inference compared to Baseline or BinaryDM:
> $$
> \text{Baseline:} w'=\sigma_1 w_1^{bi}
> $$
>
> $$
> \text{High-Order:} w'=\sigma_1 w_1^{bi}+\sigma_1 w_2^{bi}
> $$
>
> This at least doubles the parameter count and OPs. Additionally, although multiple sets of bases in higher-order methods are expected to be processed in parallel during inference, we found in our research that, to date, there has not been any implementation of this, making them computationally less efficient.
>
> For actual hardware, we implemented convolution and linear layers unit by unit to estimate the overall model, utilizing the general deployment library Larq[1] on a Qualcomm Snapdragon 855 Plus to test the actual runtime efficiency of the aforementioned single convolution. Since the current deployment libraries do not support direct computation for W1A4, we used a combined approach to achieve it via W1A1. Specifically, for the W1A4 operator, since there is no existing 4-bit activation implementation, we decompose the activation as follows:
>
> $k \cdot a^{4bit} = 4k \cdot {b_a}_1^{1bit} + 2k \cdot {b_a}_2^{1bit} + k \cdot {b_a}_3^{1bit} + \frac{1}{2}k \cdot {b_a}_4^{1bit} - \frac{1}{2}k$
>
> Where,
>
> - $k$ is the scaling factor of fp32 activation.
> - $a^{4bit} \in \{-8, -7, -6, -5, -4, -3, -2, -1, 0, 1, 2, 3, 4, 5, 6, 7\}$
> - ${b_a}_{i}^{1bit} \in \{-1, 1\}, i \in \{1, 2, 3, 4\}$
>
> As a result, the computation of 1-bit weights with int4 can be straightforwardly decomposed into the computation of 1-bit weights with 4 1-bit activations and one bias term ($\frac{1}{2}k$), based on the W1A1 operator provided by Larq, with the addition of limited arithmetic operations.
>
> The runtime results for a single inference are summarized in the table below. Due to limitations of the deployment library and hardware, Baseline/BinaryDM achieved a 4.62x speedup, while High-Order only achieved an 3.11x speedup.
>
> With further hardware support for binary operations, BinaryDM is expected to achieve performance closer to the theoretical OPs calculations (15.2x), further widening the gap between its implementation and that of high-order methods.
>
> |      Method       | #Bits |   Size   | Theoretical OPs($\times10^9$) | Runtime($\mu s$/convolution) |
> | :---------------: | :---: | :------: | :---------------------------: | ---------------------------: |
> |        FP         | 32/32 |  1045.4  |             96.0              |                     176371.0 |
> |    High-Order     |  2/4  |   70.2   |             12.6              |                      56657.5 |
> | Baseline/BinaryDM |  1/4  | **35.8** |            **6.3**            |                  **38174.2** |

---

> > ### Author Response · Authors · 2024-11-21
> > **Response to Reviewer riMU**
> >
> > [Supplementary] The corresponding decoding table is as follows:
> >
> > | $a^{4bit}$ | ${b_a}_{1}^{1bit}$ | ${b_a}_{2}^{1bit}$ | ${b_a}_{3}^{1bit}$ | ${b_a}_{4}^{1bit}$ |
> > | :--------: | :----------------: | :----------------: | :----------------: | :----------------: |
> > |     -8     |         -1         |         -1         |         -1         |         -1         |
> > |     -7     |         -1         |         -1         |         -1         |         1          |
> > |     -6     |         -1         |         -1         |         1          |         -1         |
> > |     -5     |         -1         |         -1         |         1          |         1          |
> > |     -4     |         -1         |         1          |         -1         |         -1         |
> > |     -3     |         -1         |         1          |         -1         |         1          |
> > |     -2     |         -1         |         1          |         1          |         -1         |
> > |     -1     |         -1         |         1          |         1          |         1          |
> > |     0      |         1          |         -1         |         -1         |         -1         |
> > |     1      |         1          |         -1         |         -1         |         1          |
> > |     2      |         1          |         -1         |         1          |         -1         |
> > |     3      |         1          |         -1         |         1          |         1          |
> > |     4      |         1          |         1          |         -1         |         -1         |
> > |     5      |         1          |         1          |         -1         |         1          |
> > |     6      |         1          |         1          |         1          |         -1         |
> > |     7      |         1          |         1          |         1          |         1          |
> >
> > [1] "LarQ". https://github.com/larq/larq
> >
> > We have included this part of the discussion in the revised manuscript's appendix, starting from Line 993, as well as in Table 16.

---

> > > ### Author Response · Authors · 2024-11-21
> > > **Response to Reviewer riMU**
> > >
> > > > **Q2.1.** Without theoretical nor empirical analysis regarding the location selection, I am concerned about how the authors conclude on applying the proposed approach on the first and last six layers of the models.
> > >
> > > **A2.1.** Our ultimate goal is to use EBB to enhance the learning capacity in the early stages as a transitional mechanism to achieve optimal optimization results. The clear advantage of this approach is that it strengthens early-stage representational capacity. However, such transitional enhancement comes with costs, and an inadequate transition strategy could lead to suboptimal outcomes. A natural idea is to add EBB only to the critical positions that most require early-stage learning enhancement. This allows for information enhancement while avoiding the additional burden and potential risks of large-scale transitional adjustments. As stated in Line 256 and Line 799 of the original manuscript (now Line 253 and Line 886 in the revised manuscript in the revised manuscript), applying EBB to regions with a large number of parameters but low sensitivity to binarization could lead to suboptimal optimization stability, making the precision worse compared to models where EBB is applied only to certain parts. Therefore, we specified applying EBB to the first and last six layers, as these sections have a relatively small parameter count (less than 15% of the total) and their proximity to the input and output structures makes them crucial for the generative performance of the DM.
> > >
> > > We clarified in Table 7 of the original manuscript (now Table 10 in the revised manuscript) and the following content that partial application of EBB achieves the best performance. Specifically, we have added more experimental results with finer-grained application of EBB, where applications of EBB. Here, the Head and Tail Parts value '3/6/9/...' represents EBB being applied to the first and last 3/6/9/... TimestepEmbedBlocks of the DM. Our original approach, Head and Tail Parts (6), demonstrated the highest accuracy. Additionally, while Head and Tail Parts (12) achieves lower training loss in the first 1000 iterations compared to Head and Tail Parts (6), its weaker transition to full weight binarization results in higher loss at 100K iterations.
> > >
> > > | Head and Tail Parts | Central Parts | \#Bits | FID$\downarrow$ | sFID$\downarrow$ | Prec.$\uparrow$ | Rec.$\uparrow$ |
> > > | :-----------------: | :-----------: | :----: | :-------------: | :--------------: | :-------------: | :------------: |
> > > |   0 |   0   |  1/32  |      8.02       |    12.81     |    64.83   |     33.12      |
> > > |    3   |       0       |  1/32  |      7.20       |      12.27       |      65.62      |     34.98      |
> > > |        **6**        |     **0**     |  1/32  |    **6.99**     |    **12.15**     |    **67.51**    |   **36.80**    |
> > > |          9          |       0       |  1/32  |      7.10       |      12.22       |      65.41      |     36.42      |
> > > |         12          |       1       |  1/32  |      7.10       |      12.29       |      66.41      |     34.54      |
> > >
> > > | Head and Tail Parts | Central Parts | \#Bits | FID$\downarrow$ | sFID$\downarrow$ | Prec.$\uparrow$ | Rec.$\uparrow$ |
> > > | :-----------------: | :-----------: | :----: | :-------------: | :--------------: | :-------------: | :------------: |
> > > |          0          |       0       |  1/32  |      8.02       |      12.81       |      64.83      |     33.12      |
> > > |          3          |       0       |  1/32  |      7.20       |      12.27       |      65.62      |     34.98      |
> > > |        **6**        |     **0**     |  1/32  |    **6.99**     |    **12.15**     |    **67.51**    |   **36.80**    |
> > > |          9          |       0       |  1/32  |      7.10       |      12.22       |      65.41      |     36.42      |
> > > |         12          |       1       |  1/32  |      7.10       |      12.29       |      66.41      |     34.54      |
> > >
> > > We have included this part of the discussion in the revised manuscript's appendix, starting from Line 892, as well as in Tables 9 and 10.
> > >
> > > > **Q2.2.** Does the proposed method can be applied out-of-the-box on other architectures without applying exhaustive search on which layers should be applied?
> > >
> > > **A2.2.** We declare that EBB is universal across various DM architectures and can provide consistent benefits.
> > >
> > > Specifically, wherever EBB is applied, it consistently brings stable accuracy gains compared to the vanilla scheme (also under the premise of using LRM), as shown in the table from reply A2.1 (with minimal improvements, the FID decreases from 8.02 to 7.20), highlighting its robustness in applications. As demonstrated in our original manuscript, we also applied the same setup—applying EBB only to the first and last halves of the model—in both DDIM (Table 1) and LDM-8 (Table 2), and still achieved good binarization results. Therefore, we believe that EBB can be applied out-of-the-box and will always bring positive improvements to the training of binary DM.

---

> > > > ### Author Response · Authors · 2024-11-21
> > > > **Response to Reviewer riMU**
> > > >
> > > > > **Q2.3.** This makes me even more concerned regarding the generality of the proposed method is the fact that the proposed method is computationally intensive since it applies quantization-aware training that requires the computation of the covariance matrix of the binarized and full-precision representation.
> > > >
> > > > **A2.3.** We clarify that the covariance matrix is used in LRM to compute the training loss and is unrelated to EBB. It does not affect inference efficiency, and its computation during training is minimal.
> > > >
> > > > Specifically, when using PCA for dimensionality reduction in LRM, the covariance matrix needs to be computed first, after which feature reduction is performed. As we mentioned in the original manuscript, the covariance matrix is permanently saved after its first computation and does not need to be recalculated during training. Our design primarily considered two points:
> > > >
> > > > - Line 323 points out that the computational cost of PCA dimensionality reduction is relatively high.
> > > > - Line 351 notes that a fixed transformation matrix might aid in optimization stability.
> > > >
> > > > Therefore, we assumed that a single batch of statistical data would be sufficiently representative, and, considering the cost, performed the PCA operation only once, permanently saving the transformation matrix.
> > > >
> > > > Additionally, LRM is applied to the activation features and the limited number (usually 33) of TimestepEmbedModule outputs, which incurs minimal cost and does not introduce significant additional computation during backpropagation due to EBB's design. As a result, the actual training time of BinaryDM is nearly the same as that of the Baseline.
> > > >
> > > > > **Q2.4.** This introduces significant limitations on searching the layers where the application of the proposed method could be beneficial, hindering its generalization ability.
> > > >
> > > > **A2.4.** We clarify that we did not perform any searching, but instead directly specified the application of EBB to the first and last halves of the layers. As clarified in our previous response, we believe that EBB provides stable and significant benefits in improving the accuracy of binary DM, and that the overall training scheme of BinaryDM is computationally efficient and generalizable in its application.

---

> > > > > ### Author Response · Authors · 2024-11-21
> > > > > **Response to Reviewer riMU**
> > > > >
> > > > > > **Q3.1.** The notation used in some cases is weak. For example, it is not clear whether Equation 6 is applied channel-wise.
> > > > >
> > > > > **A3.1.** We simplified the notation before writing Equation 6, where $w\in \mathbb{R}^{CHW}$ only refers to one convolutional kernel, so it is naturally channel-wise. This can also be verified in the source code in the Supplementary Material we submitted.
> > > > >
> > > > > > **Q3.2.** Additionally, in the case of the trainable parameter σ1 the domain is missing, raising the question whether the sigma can change the polarity of the binarized parameter or not.
> > > > >
> > > > > **A3.2.** The architecture of $\sigma_1$ is identical to that of traditional binary networks and is generally not considered to undergo inversion. In our actual experimental observations, it has not inverted.
> > > > >
> > > > > > **Q3.3.** Additionally, it is not clear in the text if the σ1 parameter is discarded after the training or if it is used also during the inference.
> > > > >
> > > > > **A3.3.** As shown in Equation 10 of the manuscript, $\sigma_1$ needs to be retained and involved in the computation during the inference phase.
> > > > >
> > > > > > **Q3.4.** Furthermore, using the same notation to denote different parameters as in the cases of Equations 9 and 2 can be misleading. Finally, ϵt is not properly defined in the text.
> > > > >
> > > > > **A3.4.** Thank you for your suggestion. Although we made a distinction regarding bolding, this may still lead to unnecessary misunderstandings.
> > > > >
> > > > > We have now standardized the regularization hyperparameter $\mu$ in EBB formulas to $\tau$:
> > > > > $$
> > > > > L_\text{EBB}=\tau\frac{1}{N}\sum_{i=1}^{N} \sigma_2^i
> > > > > $$
> > > > > We have also updated the intermediate representation $\epsilon$ used in Equation 13 of LRM to $\varepsilon$, for example:
> > > > > $$
> > > > > \mathcal{R}_i^{fp}(x_t, t)= {\hat\varepsilon_θ}_i^{fp}(x_t, t) E_i^{\lceil \frac{c}{K} \rfloor}
> > > > > $$
> > > > > The $\epsilon_t$ in Equation 4 now maintains the same practical meaning as the notation $\epsilon$ in Equation 14 of DDPM [1].
> > > > >
> > > > > [1] Ho, et al. Denoising Diffusion Probabilistic Models.
> > > > >
> > > > > We have incorporated this discussion into the revised manuscript at Lines 233, 291, 305, 320, and other relevant sections.
> > > > >
> > > > > > **Q4.** Although the most interesting part of the proposed method is the effective bit reduction down to 1W4A, the ablation study, which evaluates the different parts of the proposed method, is conducted only for 1W32A. I strongly recommend the authors to also include an ablation study for 1W4A, as it is the most impressive achievement of the proposed method.
> > > > >
> > > > > **A4.** Thank you for your suggestion. Our ablation experiment results on 1W4A also demonstrate the effectiveness of the two components, EBB and LRM, individually.
> > > > >
> > > > > Specifically, as shown in the table below, when EBB was added alone, the generative performance of the binary DM improved significantly, with the FID decreasing from 10.87 to 8.53. After adding LRM, the FID further decreased to 7.74, clearly illustrating the effectiveness of their synergistic effect.
> > > > >
> > > > > The reason we only presented the ablation experiments on 1W32A in the original manuscript is that we wanted to eliminate any potential impact of activation quantization on the ablation results: as the highlight of BinaryDM lies in the achievement of weight binarization for DM, the activation quantization method used was always a naive scheme without additional complex techniques.
> > > > >
> > > > >
> > > > > | Method              | \#Bits | FID$\downarrow$ | sFID$\downarrow$ | Prec.$\uparrow$ | Rec.$\uparrow$ |
> > > > > | ------------------- | :----: | --------------: | ---------------: | --------------: | -------------: |
> > > > > | Vanilla             |  1/4   |           10.87 |            15.46 |           64.05 |          26.50 |
> > > > > | +EBB                |  1/4   |            8.53 |            11.99 |           62.94 |          30.78 |
> > > > > | **+LRM (BinaryDM)** |  1/4   |        **7.74** |        **10.80** |       **64.71** |      **32.98** |
> > > > >
> > > > > We have added this part of the discussion to the revised manuscript's appendix at Line 856, as well as in Table 7.

---

> > > > > > ### Comment · Reviewer_riMU · 2024-11-23
> > > > > >
> > > > > > I would like to thank the authors for answering my question. I will update the score accordingly.
> > > > > >
> > > > > > Just a follow-up question, you mention on the text that: “Since the computational cost of obtaining the transformation matrix E^{c/K} in LRM is significantly expensive, we compute the matrix by the first batch of input and keep it fixed during the training process. The fixed mapping between representations is also beneficial to the optimization of binarized DMs from a stability perspective, as updates to the transformation matrix could significantly alter the direction of binarized optimization, which would be disastrous for DMs with high demands for representation capacity and optimization stability”. Can the authors elaborate more on this? How does the calculation of the transformation matrix only in the first batch ensure the stability of the optimization process?

---

> > > > > > > ### Author Response · Authors · 2024-11-23
> > > > > > > **Further Response to Reviewer riMU**
> > > > > > >
> > > > > > > Thank you very much for recognizing our work and the responses we have provided!
> > > > > > >
> > > > > > > Here, we offer further clarification regarding your concerns about the stability of LRM.
> > > > > > >
> > > > > > > > **Q5.** Just a follow-up question, you mention on the text that: “Since the computational cost of obtaining the transformation matrix E^{c/K} in LRM is significantly expensive, we compute the matrix by the first batch of input and keep it fixed during the training process. The fixed mapping between representations is also beneficial to the optimization of binarized DMs from a stability perspective, as updates to the transformation matrix could significantly alter the direction of binarized optimization, which would be disastrous for DMs with high demands for representation capacity and optimization stability”. Can the authors elaborate more on this? How does the calculation of the transformation matrix only in the first batch ensure the stability of the optimization process?
> > > > > > >
> > > > > > > **A5**: We clarify that we treat the first batch of samples as a random sampling of the entire input space to construct the projection matrix $E_i^{\lceil \frac{c}{K} \rfloor}$, representing the mapping of corresponding features from the entire input space to the low-dimensional space. Our analysis and empirical results demonstrate that maintaining consistency in the projection direction throughout multiple optimization steps helps avoid the interference in optimization caused by changes in the projection matrix during the process.
> > > > > > >
> > > > > > > Specifically, LRM employs a dimensionality reduction matrix, $E_i^{\lceil \frac{c}{K} \rfloor}$, to project both full-precision and binarized features from each iteration into a low-rank space, enabling the binarized DM to better align with the full-precision model in this reduced space. Notably, $E_i^{\lceil \frac{c}{K} \rfloor}$​ originates from the transformation of the full-precision feature space. While this approach effectively preserves the critical information of full-precision features, applying the same projection to binarized features could significantly alter their properties, while also causing drastic shifts in the entire low-rank space.
> > > > > > >
> > > > > > > In such cases, frequent updates to the projection matrix may result in instability. Due to the dynamic nature of DMs, frequent updates can cause dramatic changes to the projection matrix, which, in turn, could destabilize the binarized DM's feature outputs. This instability makes it difficult for the binarized DM to align stably within the constantly evolving low-rank space, as both the optimization target (the low-rank space based on full-precision features) and the optimization process (the projection of binarized features) are continuously changing.
> > > > > > >
> > > > > > > To validate the above analysis, as explained in our response `A5` to `Reviewer F7Tj`'s question `Q5`, we conducted experiments where the dimensionality reduction matrix $E_i^{\lceil \frac{c}{K} \rfloor}$ was updated every 100 iterations. As shown in the table below, while using LRM consistently improves performance (e.g., reducing FID from 7.39 to 7.11/6.99), the approach of initializing the matrix once and retaining it throughout yields the highest accuracy. This corroborates the analysis in Line 351 of the manuscript and the perspective you cited, which states that fixing the dimensionality reduction matrix without frequent updates is more beneficial for stable optimization.
> > > > > > >
> > > > > > > | Update Frequency (/iter) | \#Bits | FID$\downarrow$ | sFID$\downarrow$ |
> > > > > > > | ------------------------ | :----: | --------------: | ---------------: |
> > > > > > > | 0 (w/o LRM)              |  1/32  |            7.39 |            12.34 |
> > > > > > > | 100                      |  1/32  |            7.11 |            12.23 |
> > > > > > > | **$\infty$ (BianryDM)**  |  1/32  |        **6.99** |        **12.15** |

---

> > > > > > > > ### Comment · Reviewer_riMU · 2024-11-25
> > > > > > > >
> > > > > > > > Thanks the authors for taking effort to answer my concerns!
> > > > > > > >
> > > > > > > > Please make sure that the additional analysis provided in the rebuttal will be included in the paper.

---

> > > > > > > > > ### Author Response · Authors · 2024-11-26
> > > > > > > > > **Response to Reviewer riMU**
> > > > > > > > >
> > > > > > > > > Dear Reviewer riMU,
> > > > > > > > >
> > > > > > > > > Thank you once again for your recognition of our work and responses!
> > > > > > > > >
> > > > > > > > > Your discussion has greatly improved our work, and we will continue to add the discussed content in the upcoming revised version.
> > > > > > > > >
> > > > > > > > > Best regards,
> > > > > > > > >
> > > > > > > > > Authors of Paper 1841

---

### Official Review · Reviewer_A5R1 · 2024-11-03

**Soundness:** 3
**Presentation:** 3
**Contribution:** 3
**Rating:** 8
**Confidence:** 5

**Summary:**

This paper introduces BinaryDM achieving weight binarization in diffusion models while mitigating accuracy degradation. BinaryDM tackles the challenges of information loss and optimization difficulties inherent in binarization through two key techniques: Evolvable-Basis Binarizer (EBB) and Low-rank Representation Mimicking (LRM). EBB employs a multi-basis structure during training, allowing for a smooth transition to a single-basis binary representation for inference, thus enhancing representational capacity. LRM, on the other hand, facilitates the stable convergence of binarized DMs by mimicking the representations of a full-precision counterpart in a low-rank space, addressing the optimization direction ambiguity caused by extreme discretization.

**Strengths:**

1. Existing binarization works benefit a lot from exploiting residual connections which holds full precision information in the connection it self without information loss. This method exploit this benefit to the initial state of binarization to start with good initial state for lowering accuracy degradation without changing any network structure in the final state.
2. BinaryDM demonstrates superior accuracy and efficiency compared to existing binarization and low-bit quantization methods.

**Weaknesses:**

Please see the questions below.

**Questions:**

1. While BinaryDM shows good result in diffusion models, there are many works that prove good result on other models like ReActNet [1] and InstaBNN [2] (There are much more existing works). Since there are not many existing works in diffusion model quantization, the results in this paper is not enough to prove the true superiority of BinaryDM.
2. On the Evolvable Basis Binarizer (Equation 5), is there any reason the authors used “\sigma_{II} sign(w - \sigma_1 sign(w))” instead of “\sigma_{II} (w - \sigma_1 sign(w))” in second term? In this reviewer’s intuitive thinking, not utilizing sign function in second term would give more favorable initial state.
3. Details: Are the number of training iterations of BinaryDM and existing works same? Also curious if the learning rate or schedulers are used same as in existing works.

[1] Zechun Liu, Zhiqiang Shen, Marios Savvides, and Kwang-Ting Cheng. Reactnet: Towards precise binary neural network with generalized activation functions. ECCV, 2020

[2] Changhun Lee, Hyungjun Kim, Eunhyeok Park, and Jae-Joon Kim, INSTA-BNN: Binary Neural Network with INSTAnce-aware Threshold, ICCV, 2023

---

> ### Author Response · Authors · 2024-11-21
> **Response to Reviewer A5R1**
>
> We thank you for your positive feedback and comments. We respond to the concerns below:
>
> > **Q1.** While BinaryDM shows good result in diffusion models, there are many works that prove good result on other models like ReActNet [1] and InstaBNN [2] (There are much more existing works). Since there are not many existing works in diffusion model quantization, the results in this paper is not enough to prove the true superiority of BinaryDM.
>
> **A1.** Thank you for your suggestion. Following your advice, we have added experiments comparing the most advanced general binarization methods, ReActNet [1] and InstaBNN [2], with the latest binarized DM approach, BI-DiffSR [3]. The results show that BinaryDM, as a weight binarization method for DM designed specifically for generative tasks, still achieves the highest accuracy in image generation, further highlighting the effectiveness of BinaryDM.
>
> Specifically, we followed your suggestion to include experiments using ReActNet and InstaBNN. ReActNet, which improves RSign and PReLU, has long been widely recognized as an effective binarization method in the binary domain. InstaBNN further deepens the understanding of thresholding by incorporating input statistics for dynamic adjustments, and has been validated on CNN-based classification tasks, making it one of the most advanced binarization methods. In addition to these two methods, we also included BI-DiffSR, a recently published binarization method for DM applied to image super-resolution tasks. Moreover, in our original manuscript, such as in Table 2, we had compared with several advanced QAT quantization techniques for DM.
>
> The experimental comparison results with these methods are shown in the table below. BinaryDM, as the only weight binarization method specifically designed for generative DM, still demonstrates the best generation performance. General advanced binarization methods, such as RSign and PReLU, struggle to be applied effectively to the respective scenarios, while advanced DM QAT methods have difficulty adapting to the 1-bit extreme compression environment or performing well in generative tasks that start from initial noise.
>
> | Method       | \#Bits | FID$\downarrow$ | sFID$\downarrow$ | Prec.$\uparrow$ | Rec.$\uparrow$ |
> | ------------ | :----: | --------------: | ---------------: | --------------: | -------------: |
> | Q-Diffusion  |  4/4   |          427.46 |           277.22 |            0.00 |           0.00 |
> | EfficientDM  |  4/4   |           10.60 |                - |               - |              - |
> | LSQ          |  2/4   |           12.95 |            12.79 |           55.97 |          34.30 |
> | Baseline     |  1/4   |           10.87 |            15.46 |           64.05 |          26.50 |
> | TDQ          |  1/4   |           11.28 |            12.80 |           55.14 |          27.32 |
> | ReActNet     |  1/4   |           10.23 |            13.02 |           61.43 |          29.68 |
> | Q-DM         |  1/4   |            9.99 |            11.96 |           57.62 |          29.30 |
> | INSTA-BNN    |  1/4   |            9.42 |            12.39 |           60.05 |          31.08 |
> | BI-DiffSR    |  1/4   |            8.58 |            11.81 |           62.61 |          30.86 |
> | **BinaryDM** |  1/4   |        **7.74** |        **10.80** |       **64.71** |      **32.98** |
>
> [1] Zechun Liu, Zhiqiang Shen, Marios Savvides, and Kwang-Ting Cheng. Reactnet: Towards precise binary neural network with generalized activation functions. ECCV, 2020
>
> [2] Changhun Lee, Hyungjun Kim, Eunhyeok Park, and Jae-Joon Kim, INSTA-BNN: Binary Neural Network with INSTAnce-aware Threshold, ICCV, 2023
>
> [3] Chen, et al. Binarized Diffusion Model for Image Super-Resolution. NeurIPS 2024.
>
> We have incorporated this part of the discussion into Table 2 of the revised manuscript's main text and Line 820 of the appendix.

---

> > ### Author Response · Authors · 2024-11-21
> > **Response to Reviewer A5R1**
> >
> > > **Q2.** On the Evolvable Basis Binarizer (Equation 5), is there any reason the authors used “$\sigma_{II} sign(w - \sigma_1 sign(w))$” instead of “$\sigma_{II} (w - \sigma_1 sign(w))$” in second term? In this reviewer’s intuitive thinking, not utilizing sign function in second term would give more favorable initial state.
> >
> > **A2.** Thank you for the insights you provided. However, through analysis and experiments, we have found that the existing EBB approach is still well-suited for optimizing binarized DM.
> >
> > Specifically, while this is a very natural idea—as it would provide stronger fitting ability in the early stages—it is not beneficial for achieving a fully binarized final model through optimization. This is because it would lead to a significant imbalance between the representations of the initial and final stages, causing the second term corresponding to $\sigma_{II}$ to dominate due to its excessive information extraction capability, which in turn hinders the subsequent optimization of $\sigma_1 \text{sign}(w)$.
> >
> > The additional experimental results we present in the table below show that this idea did not achieve the optimal generation performance. At the same time, we observed the value of $\sigma_{II}$ at the end of the first phase of EBB training. We found that, unlike the original approach where regularization would force it to converge to nearly zero, $\sigma_{II}$ still had a relatively large value, which indicates that this idea indeed hindered the natural evolution from multiple bases to a single base, and reduced the effectiveness of EBB as a method designed to enhance learning ability.
> >
> >
> > | Method                 | \#Bits | FID$\downarrow$ | sFID$\downarrow$ | Prec.$\uparrow$ | Rec.$\uparrow$ |
> > | ---------------------- | :----: | --------------: | ---------------: | --------------: | -------------: |
> > | Baseline               |  1/4   |           10.87 |            15.46 |           64.05 |          26.50 |
> > | no sign in second term |  1/4   |            8.44 |            13.00 |           62.68 |          30.12 |
> > | **BinaryDM**           |  1/4   |        **7.74** |        **10.80** |       **64.71** |      **32.98** |
> >
> > > **Q3.** Details: Are the number of training iterations of BinaryDM and existing works same? Also curious if the learning rate or schedulers are used same as in existing works.
> >
> > **A3.** We have kept the hyperparameter settings consistent and tried to align them with those explicitly mentioned in other works.
> >
> > Specifically, taking LDM-4 on LSUN-Bedrooms as an example, we used three different settings for the training iterations in our experiments:
> >
> > - PTQ: Only Q-Diffusion.
> > - Efficient QAT: Only EfficientDM. We found that further increasing the number of iterations did not yield additional gains for LoRA-based EfficientDM.
> > - Other QATs: For example, BinaryDM, Baseline, Q-DM, TDQ. We trained uniformly for 200K iterations, with a learning rate set to 2.0e-05, and used schedulers that were consistent with the naive full-precision LDM. More details can be found in section B.1 of the appendix and in the source code provided in the Supplementary Material of our original submission.
> >
> > Therefore, we conducted our experiments with consistent parameters to ensure fairness. Additionally, although we followed existing works as much as possible in terms of experimental settings (e.g., using the same batch size as EfficientDM), since other QAT works have not fully disclosed their parameters or source code, and BinaryDM is the first to apply QAT to binarized DM, we made every effort to optimize our parameter settings starting from a better baseline after multiple trials.

---

> > > ### Comment · Reviewer_A5R1 · 2024-11-25
> > >
> > > Thanks for your clear explanation for the concerns. Especially, thanks for adding comparison results for Answer1. I think all my concerns are resolved.
> > >
> > > Please include our discussion here in appendix afterwards.

---

> > > > ### Author Response · Authors · 2024-11-26
> > > > **Response to Reviewer A5R1**
> > > >
> > > > Dear Reviewer A5R1,
> > > >
> > > > Thank you for your recognition of our work and responses!
> > > >
> > > > Through the concerns you raised regarding our work, we have further clarified the motivation and effectiveness of BinaryDM. Based on the current revised version, we will continue to supplement the discussed content in the subsequent revision.
> > > >
> > > > Best regards,
> > > >
> > > > Authors of Paper 1841

---

### Official Review · Reviewer_F7Tj · 2024-11-03

**Soundness:** 3
**Presentation:** 3
**Contribution:** 3
**Rating:** 6
**Confidence:** 4

**Summary:**

The paper introduces BinaryDM, a novel approach for accurate weight binarization of diffusion models (DMs) to achieve efficient deployment and inference on edge devices. The authors identify that binarizing DMs is challenging due to severe accuracy degradation caused by limited representational capacity and optimization difficulties. To address these issues, they propose two key techniques: 1)Evolvable-Basis Binarizer (EBB): This method enhances the initial representation capacity by starting with multiple binary bases and learnable scalars, which are gradually regularized to evolve into efficient single-basis binarization during training. EBB is selectively applied to crucial parts of the DM architecture to retain stability and reduce training overhead. 2)Low-rank Representation Mimicking (LRM): This technique improves the optimization process by aligning the low-rank representations of the binarized DM with those of a full-precision counterpart, mitigating the optimization direction ambiguity caused by fine-grained alignment. The authors conduct extensive experiments on various datasets and demonstrate that BinaryDM achieves significant accuracy improvements over state-of-the-art quantization methods under ultra-low bit-widths, while also providing substantial efficiency gains in terms of model size and computational operations.

**Strengths:**

1. The paper addresses the challenging problem of binarizing diffusion models. The proposed methods, EBB and LRM, are novel and tailored to address specific issues in binarizing DMs.
2. The authors provide a thorough experimental evaluation of their methods across multiple datasets and models. The results demonstrate that BinaryDM outperforms existing quantization methods under ultra-low bit-width settings, validating the effectiveness of the proposed techniques.
3. The paper is generally well-written and organized. The motivation behind the proposed methods is clearly explained, and the technical details are presented with sufficient depth. Figures and tables effectively illustrate the concepts and results.
4. Achieving efficient and accurate binarization of DMs is important for deploying these models on resource-constrained devices. The substantial efficiency gains demonstrated by BinaryDM highlight its potential impact in practical applications.

**Weaknesses:**

1. The Evolvable-Basis Binarizer (EBB) is selectively applied only to the first and last six layers of the diffusion model, covering only a small portion of the total parameters. This limited application scope raises concerns regarding the method’s generality and effectiveness throughout the network. Although the paper suggests that these head and tail layers are critical and parameter-sparse, it lacks a thorough justification or empirical evidence for this choice. Additionally, the middle layers, which remain untouched by EBB, may contribute significantly to the model’s performance and could also benefit from the method. By limiting EBB to specific layers, the paper potentially misses opportunities to enhance the representational capacity of the entire network. A more comprehensive exploration of applying EBB across different layers, including ablation studies, could provide insights into its most effective usage.
2. The paper introduces the Evolvable-Basis Binarizer (EBB) and Low-rank Representation Mimicking (LRM) to address challenges in training binarized diffusion models. However, it lacks a thorough analysis of training stability and convergence behavior. Binarized models are notoriously difficult to optimize due to the discrete nature of weights and activations, which can lead to issues like gradient mismatch and convergence to poor local minima. The paper would benefit from a detailed investigation into how EBB and LRM specifically improve training dynamics.
3. The paper compares BinaryDM primarily with baseline binarization methods and some low-bit quantization techniques like LSQ and Q-Diffusion. However, it does not thoroughly benchmark against the latest state-of-the-art quantization and binarization methods specifically designed for diffusion models or other generative models. Recent advances might offer competitive performance. A more comprehensive comparison would provide a clearer picture of where BinaryDM stands.
4. The paper focuses on empirical results but lacks a theoretical analysis of why the proposed methods improve performance. Providing theoretical insights or analysis could enhance the understanding of the underlying mechanisms.
5. Some technical details, particularly in the description of LRM and the training procedure, could be elaborated further to improve clarity. For instance, the choice of hyperparameters, how the low-rank projection matrices are computed and fixed, and the stability considerations during training.

**Questions:**

1. Can the authors provide more justification for the selective application of EBB to only the first and last six layers of the DM architecture? How critical is this choice, and how does it affect performance and training stability?
2. While the paper focuses on diffusion models, are there other binarization methods from other domains (e.g., binarized neural networks in classification tasks) that could be adapted for DMs? How does BinaryDM compare with such methods?

---

> ### Author Response · Authors · 2024-11-21
> **Response to Reviewer F7Tj**
>
> We are deeply grateful for your support of our work, and we provide detailed responses to your comments as follows:
>
> > **Q1.** The Evolvable-Basis Binarizer (EBB) is selectively applied only to the first and last six layers of the diffusion model ...
>
> **A1.** Thank you for your suggestion. Our ultimate goal is to use EBB to enhance the learning capacity in the early stages as a transitional mechanism to achieve optimal optimization results. The clear advantage of this approach is that it strengthens early-stage representational capacity. However, such transitional enhancement comes with costs, and an inadequate transition strategy could lead to suboptimal outcomes. A natural idea is to add EBB only to the critical positions that most require early-stage learning enhancement. This allows for information enhancement while avoiding the additional burden and potential risks of large-scale transitional adjustments. As stated in Line 256 and Line 799 of the original manuscript (now Line 253 and Line 886 in the revised manuscript), applying EBB to regions with a large number of parameters but low sensitivity to binarization could lead to suboptimal optimization stability, making the precision worse compared to models where EBB is applied only to certain parts. Therefore, we specified applying EBB to the first and last six layers, as these sections have a relatively small parameter count (less than 15% of the total) and their proximity to the input and output structures makes them crucial for the generative performance of the DM.
>
> Our empirical research also shows that the current BinaryDM approach, which applies EBB only to the first and last six layers, yields the best results. Specifically, we briefly explored this in Table 7 of the original manuscript (now Table 10 in the revised manuscript), where we tested the performance of models with no application/partial application/full application of EBB. The results showed that partial application yielded the best performance. To further address your question, we have conducted additional experiments with finer-grained applications of EBB. Here, the Head and Tail Parts value '3/6/9/...' represents EBB being applied to the first and last 3/6/9/... TimestepEmbedBlocks of the DM:
>
> - The first table measures training loss across different iterations (1e1/2/3/4/5), demonstrating that our original configuration (Head and Tail Parts (6)) achieves the best convergence (with a minimal loss of 0.130 at 100K iterations).
> - The second table indicates that our original approach also achieves the highest accuracy.
>
> These results further validate the conclusions we presented in Lines 256 and 799 of the original manuscript. Specifically, applying EBB to regions with high parameter counts but lower sensitivity to binarization can lead to suboptimal optimization stability, resulting in worse performance compared to applying EBB selectively. Additionally, while Head and Tail Parts (12) achieves lower training loss in the first 1000 iterations compared to Head and Tail Parts (6), its weaker transition to full weight binarization results in higher loss at 100K iterations.
>
> Notably, regardless of where EBB is applied, it consistently provides accuracy improvements over the vanilla approach (under the same LRM assumptions). Even the minimal improvement reduces FID from 8.02 to 7.20, highlighting the robustness of EBB in practical applications.
>
> We have included this part of the discussion in the revised manuscript's appendix, starting from Line 892, as well as in Tables 9 and 10.
>
> | Head and Tail Parts | Central Parts | \#Bits | 1e1   | 1e2   | 1e3   | 1e4       | 1e5       |
> | :-----------------: | :-----------: | ------ | ----- | ----- | ----- | --------- | --------- |
> |          0          |       0       | 1/32   | 0.335 | 0.291 | 0.268 | 0.202     | 0.141     |
> |          3          |       0       | 1/32   | 0.335 | 0.263 | 0.230 | 0.184     | 0.138     |
> |        **6**        |     **0**     | 1/32   | 0.332 | 0.238 | 0.199 | **0.178** | **0.130** |
> |          9          |       0       | 1/32   | 0.331 | 0.223 | 0.201 | 0.183     | 0.133     |
> |         12          |       1       | 1/32   | 0.331 | 0.225 | 0.197 | 0.188     | 0.136     |
>
> | Head and Tail Parts | Central Parts | \#Bits | FID$\downarrow$ | sFID$\downarrow$ | Prec.$\uparrow$ | Rec.$\uparrow$ |
> | :-----------------: | :-----------: | :----: | :-------------: | :--------------: | :-------------: | :------------: |
> |          0          |       0       |  1/32  |      8.02       |      12.81       |      64.83      |     33.12      |
> |          3          |       0       |  1/32  |      7.20       |      12.27       |      65.62      |     34.98      |
> |        **6**        |     **0**     |  1/32  |**6.99** | **12.15**     |    **67.51**    |   **36.80**    |
> |9|0|1/32|7.10|12.22|65.41|36.42 |
> |12 |1|  1/32  |7.10|12.29 |66.41| 34.54 |

---

> > ### Author Response · Authors · 2024-11-21
> > **Response to Reviewer F7Tj**
> >
> > > **Q2.** The paper introduces the Evolvable-Basis Binarizer (EBB) and Low-rank Representation Mimicking (LRM) to address challenges in training binarized diffusion models. However, it lacks a thorough analysis of training stability and convergence behavior. Binarized models are notoriously difficult to optimize due to the discrete nature of weights and activations, which can lead to issues like gradient mismatch and convergence to poor local minima. The paper would benefit from a detailed investigation into how EBB and LRM specifically improve training dynamics.
> >
> > **A2.** Thank you for your suggestion. EBB is designed to enhance information representation through the Evolvable-Basis Binarizer, addressing optimization challenges such as weak learning capacity and slow early convergence during the initial stages of training binary DMs, which stem from the limited representational power and structural constraints of binary weights. LRM, on the other hand, employs a dimensionality reduction matrix obtained through PCA to align binary DMs with full-precision DMs in a low-rank space, enabling effective supervision and adjustment for learnable optimization directions. Both methods are specifically designed for optimizing binary DMs, which are notoriously difficult to train, leading to binary DMs with improved learning capacity and superior generative performance.
> >
> > To further address your concerns, we analyze the convergence and stability of EBB and LRM from three perspectives: local theoretical analysis or intermediate results, training loss, and the final accuracy of trained models.
> >
> > For EBB:
> >
> > - Local Theoretical Analysis: We use the example $0<\sigma_2<\sigma_1$ to illustrate two cases:
> >
> >   - Case 1: During the early stage of training, suppose a certain latent weight $ w = \sigma_1 + \varepsilon $ ($ \varepsilon > 0 $) at a specific iteration. If the gradient update provided by STE during backpropagation is $ -\delta $ ($ \delta > \varepsilon $), then after the update, $ w' = \sigma_1 + \varepsilon - \delta $.
> >
> >     - In the Baseline approach, $ w_{\text{baseline}} $ remains at $ \sigma_1 \text{sign}(w') = \sigma_1 \text{sign}(w) = \sigma_1 $, failing to provide timely feedback for updates to the real-valued weight.
> >     - In contrast, $ w_{\text{EBB}} $ is given by $ \sigma_1 \text{sign}(w) + \sigma_2 \text{sign}(w - \sigma_1 \text{sign}(w)) = \sigma_1 + \sigma_2 $, which updates to $ \sigma_1 \text{sign}(w') + \sigma_2 \text{sign}(w' - \sigma_1 \text{sign}(w')) = \sigma_1 - \sigma_2 $, providing timely feedback for adjustment.
> >
> >   - Case 2: Near the zero point, suppose $ w' = \varepsilon $ ($ 0 < \varepsilon < \sigma_1 $) at a specific iteration. If the gradient update provided by STE is $ -\delta $ ($ \delta > \varepsilon $), then after the update, $ w' = \varepsilon - \delta $.
> >
> >     - In the Baseline approach, $ w_{\text{baseline}} $ switches from $ \sigma_1 $ to $ -\sigma_1 $, experiencing a significant change ($ 2\sigma_1 $).
> >     - For $ w_{\text{EBB}} $, the change is from $ \sigma_1 - \sigma_2 $ to $ -\sigma_1 + \sigma_2 $, which is less dramatic ($ 2(\sigma_1 - \sigma_2) $) and better aligns with the true changes in the latent weight.
> >
> >     Thus, we believe the introduction of $ \sigma_2 $ in EBB enables more stable training, particularly in the early stages. Since the training of deep neural networks is widely regarded as highly sensitive to the quality of initialization, the improvements brought by EBB are likely to have a lasting beneficial impact on training stability.
> >
> > - Training Loss. The following table samples the training loss ($L_{simple}$) across different iterations (1e1/2/3/4/5). The results indicate that EBB consistently achieves lower training loss, demonstrating its benefits for convergence.
> >
> >   | Method\Iteration | \#Bits |       1e1 |       1e2 |       1e3 |       1e4 |       1e5 |
> >   | ---------------- | :----: | --------: | --------: | --------: | --------: | --------: |
> >   | Baseline         |  1/32  |     0.388 |     0.303 |     0.277 |     0.227 |     0.158 |
> >   | **+EBB**         |  1/32  | **0.352** | **0.264** | **0.242** | **0.206** | **0.151** |
> >
> > - Final Model Accuracy. The table below presents the evaluation results on LSUN-Bedrooms. The results show that EBB consistently achieves superior accuracy, demonstrating its enhanced learning capability.
> >
> >   | Method   | \#Bits | FID$\downarrow$ | sFID$\downarrow$ | Prec.$\uparrow$ | Rec.$\uparrow$ |
> >   | -------- | :----: | --------------: | ---------------: | --------------: | -------------: |
> >   | Baseline |  1/32  |            8.43 |            13.11 |           65.45 |          29.88 |
> >   | **+EBB** |  1/32  |        **7.39** |        **12.34** |       **65.98** |      **35.84** |

---

> > > ### Author Response · Authors · 2024-11-21
> > > **Response to Reviewer F7Tj**
> > >
> > > For LRM：
> > >
> > > - Intermediate Results Analysis. As illustrated in Figure 3 of the original manuscript, LRM effectively improves the alignment of binary DM with full-precision DM. Notably, even when using $L_2-Distance$ to measure the discrepancy between full-precision intermediate features ($\epsilon^{fp}$) (now $\varepsilon^{fp}$ in the revised manuscript) and binary intermediate features ($\epsilon^{bi}$) (now $\varepsilon^{bi}$ in the revised manuscript), the loss of the binary DM optimized with LRM (blue curve) remains lower than the result obtained by directly using the $L_2-Distance$ as the optimization supervision objective (green curve). This highlights the superiority of LRM over standard distillation or non-distillation approaches in aiding training convergence.
> > >
> > > - Training Loss. The table below shows the training loss ($L_{simple}$) across different iterations (1e1/2/3/4/5). The results indicate that LRM consistently achieves lower training loss, further demonstrating its benefits for convergence.
> > >
> > >   | $L_{distil}$\Iteration | \#Bits |       1e1 |       1e2 |       1e3 |       1e4 |       1e5 |
> > >   | :--------------------: | :----: | --------: | --------: | --------: | --------: | --------: |
> > >   |           -            |  1/32  |     0.352 |     0.264 |     0.242 |     0.206 |     0.151 |
> > >   |       $L_{MSE}$        |  1/32  |     0.343 |     0.249 |     0.213 |     0.184 |     0.149 |
> > >   |       $L_{LRM}$        |  1/32  | **0.332** | **0.238** | **0.199** | **0.178** | **0.130** |
> > >
> > > - Final Model Accuracy. As shown in Table 10 of the appendix in the original manuscript  (now Table 14 in the revised manuscript), LRM outperforms both MSE and non-distillation approaches across various evaluation metrics, demonstrating its superior learning capability.
> > >
> > >   | $L_{distil}$ | \#Bits | FID$\downarrow$ | sFID$\downarrow$ | Prec.$\uparrow$ | Rec.$\uparrow$ |
> > >   | :----------: | :----: | --------------: | ---------------: | --------------: | -------------: |
> > >   |      -       |  1/32  |            7.39 |            12.34 |           65.98 |          35.84 |
> > >   |  $L_{MSE}$   |  1/32  |            7.36 |            12.76 |           62.05 |          33.64 |
> > >   |  $L_{LRM}$   |  1/32  |        **6.99** |        **12.15** |       **67.51** |      **36.80** |
> > >
> > > We have added this part of the discussion to the revised manuscript's appendix at Lines 873 and 949, as well as in Tables 8 and 13.

---

> > > > ### Author Response · Authors · 2024-11-21
> > > > **Response to Reviewer F7Tj**
> > > >
> > > > > **Q3.** The paper compares BinaryDM primarily with baseline binarization methods and some low-bit quantization techniques like LSQ and Q-Diffusion. However, it does not thoroughly benchmark against the latest state-of-the-art quantization and binarization methods specifically designed for diffusion models or other generative models. Recent advances might offer competitive performance. A more comprehensive comparison would provide a clearer picture of where BinaryDM stands.
> > > >
> > > > **A3.** Thank you for your suggestion. Following your advice, we have further supplemented our comparisons with advanced quantization strategies such as ReactNet [1], INSTA-BNN [2], and BI-DiffSR [3]. The results still indicate that BinaryDM achieves the best performance. ReactNet and INSTA-BNN are state-of-the-art general-purpose binarization methods, while BI-DiffSR is the latest publicly available binarization method applied to diffusion models (DMs) in the image super-resolution task. The additional experimental results are shown in the table below.
> > > >
> > > > In our original manuscript, such as in Table 2, we compared BinaryDM with several advanced DM quantization approaches, including EfficientDM [4], Q-DM [5], and TDQ [6], all of which are precise QAT methods. BinaryDM even surpasses higher-bit methods in certain scenarios. For example, on LSUN-Bedrooms, BinaryDM in W1A4 achieves an FID of 7.74, outperforming EfficientDM in W4A4 with an FID of 10.60.
> > > >
> > > > The table below presents the results of these methods on LSUN-Bedrooms under 4-bit activation quantization. The experiments demonstrate that BinaryDM, as a binarization method tailored for DMs performing image generation tasks, benefits significantly from its proposed EBB and LRM. These innovations enable BinaryDM to surpass other advanced methods across all metrics, highlighting its effectiveness in DM applications.
> > > >
> > > > | Method       | \#Bits | FID$\downarrow$ | sFID$\downarrow$ | Prec.$\uparrow$ | Rec.$\uparrow$ |
> > > > | ------------ | :----: | --------------: | ---------------: | --------------: | -------------: |
> > > > | Q-Diffusion  |  4/4   |          427.46 |           277.22 |            0.00 |           0.00 |
> > > > | EfficientDM  |  4/4   |           10.60 |                - |               - |              - |
> > > > | LSQ          |  2/4   |           12.95 |            12.79 |           55.97 |          34.30 |
> > > > | Baseline     |  1/4   |           10.87 |            15.46 |           64.05 |          26.50 |
> > > > | TDQ          |  1/4   |           11.28 |            12.80 |           55.14 |          27.32 |
> > > > | Reactnet     |  1/4   |           10.23 |            13.02 |           61.43 |          29.68 |
> > > > | Q-DM         |  1/4   |            9.99 |            11.96 |           57.62 |          29.30 |
> > > > | INSTA-BNN    |  1/4   |            9.42 |            12.39 |           60.05 |          31.08 |
> > > > | BI-DiffSR    |  1/4   |            8.58 |            11.81 |           62.61 |          30.86 |
> > > > | **BinaryDM** |  1/4   |        **7.74** |        **10.80** |       **64.71** |      **32.98** |
> > > >
> > > > [1] Liu, et al. Reactnet: Towards precise binary neural network with generalized activation functions. ECCV 2020.
> > > >
> > > > [2] Lee, et al. INSTA-BNN: Binary neural network with instance-aware threshold. ICCV 2023.
> > > >
> > > > [3] Chen, et al. Binarized Diffusion Model for Image Super-Resolution. NeurIPS 2024.
> > > >
> > > > [4] He, et al. Efficientdm: Efficient quantization-aware fine-tuning of low-bit diffusion models. ICLR 2024.
> > > >
> > > > [5] Li, et al. Q-dm: An efficient low-bit quantized diffusion model. NeurIPS 2023.
> > > >
> > > > [6] So, et al. Temporal Dynamic Quantization for Diffusion Models. NeurIPS 2023.
> > > >
> > > > We have incorporated this part of the discussion into Table 2 of the revised manuscript's main text and Line 820 of the appendix.

---

> > > > > ### Author Response · Authors · 2024-11-21
> > > > > **Response to Reviewer F7Tj**
> > > > >
> > > > > > **Q4.** The paper focuses on empirical results but lacks a theoretical analysis of why the proposed methods improve performance. Providing theoretical insights or analysis could enhance the understanding of the underlying mechanisms.
> > > > >
> > > > > **A4.** Thank you for your suggestion. In our response `A2` to your question `Q2`, we systematically analyzed the convergence and stability of EBB and LRM from three perspectives: local theoretical analysis or intermediate results, training loss and the final model accuracy. For reasons behind the benefits brought by certain local details, such as the choice of EBB's scope, we provided more detailed analysis and explanation in our response `A1` to your question `Q1`.
> > > > >
> > > > > We have incorporated this content into the revised version of the manuscript to make the theoretical insights of BinaryDM more thorough and comprehensive.
> > > > >
> > > > > > **Q5.** Some technical details, particularly in the description of LRM and the training procedure, could be elaborated further to improve clarity. For instance, the choice of hyperparameters, how the low-rank projection matrices are computed and fixed, and the stability considerations during training.
> > > > >
> > > > > **A5.** Here, we provide further clarification on these detailed processes and parameters and explain the efforts we have made to ensure the training stability of LRM.
> > > > >
> > > > > In section B.1 of the appendix, we outlined some of the experimental parameter settings, and these details are faithfully reflected in the code in the Supplementary Material. Below are further details:
> > > > >
> > > > > - The value of $\mu$ in Equation 9 (now $\tau$ in the revised manuscript) is typically set to 9e-2 , and the value of $\lambda$ in Equation 15 is typically set to 1e-4.
> > > > > - In the first iteration of training, we compute the dimensionality reduction matrix $E_i^{\lceil \frac{c}{K} \rfloor}$ using the first batch of inputs, utilizing Equations 11 and 12.
> > > > >
> > > > > As an additional clarification on stability, we also conducted experiments where the dimensionality reduction matrix $E_i^{\lceil \frac{c}{K} \rfloor}$ is updated every 100 iterations. As shown in the table below, while using LRM consistently yields improvements (with FID decreasing from 7.39 to 7.11/6.99), the approach of initializing the matrix once and retaining it throughout results in the highest accuracy. This further confirms the analysis made in Line 351 of the manuscript, where we stated that fixing the dimensionality reduction matrix and not updating it is more beneficial for stable optimization.
> > > > >
> > > > > | Update Frequency (/iter) | \#Bits | FID$\downarrow$ | sFID$\downarrow$ |
> > > > > | ------------------------ | :----: | --------------: | ---------------: |
> > > > > | 0 (w/o LRM)              |  1/32  |            7.39 |            12.34 |
> > > > > | 100                      |  1/32  |            7.11 |            12.23 |
> > > > > | **$\infty$ (BianryDM)**  |  1/32  |        **6.99** |        **12.15** |
> > > > >
> > > > > We have added this part of the discussion to the revised manuscript's appendix at Line 969, as well as in Table 15.
> > > > >
> > > > > > **Q6.** Can the authors provide more justification for the selective application of EBB to only the first and last six layers of the DM architecture? How critical is this choice, and how does it affect performance and training stability?
> > > > >
> > > > > **A6.** As we analyzed and validated in Section 3.2 of the original manuscript under "Location Selection" and in Appendix B.2 under "Effects of EBB", as well as in the response `A1` to your query `Q1`, applying EBB across the entire network, especially to areas with a large number of parameters but low sensitivity to binarization, may lead to suboptimal optimization stability. This is why the scheme of applying EBB only to the first and last six layers achieved the optimal FID (6.99). On the other hand, EBB consistently showed stable improvements; wherever applied, it resulted in accuracy gains, highlighting its significant effectiveness for binary DM.
> > > > >
> > > > > In addition to Section 3.2's "Location Selection" and our response `A1` to your query `Q1`, we have also further supplemented an in-depth analysis and verification of this issue starting from Line 886 in the revised manuscript.
> > > > >
> > > > > > **Q7.** While the paper focuses on diffusion models, are there other binarization methods from other domains (e.g., binarized neural networks in classification tasks) that could be adapted for DMs? How does BinaryDM compare with such methods?
> > > > >
> > > > > **A7.** As we explained in response `A3` to your query `Q3`, we have added experimental results on advanced general binarization methods such as Reactnet and INSTA-BNN, and still found that BinaryDM achieves the highest accuracy. This confirms that BinaryDM, as a weight binarization method specifically designed for DM in generative tasks, has a significant accuracy advantage due to its unique designs, EBB and LRM, in this field.
> > > > >
> > > > > In addition to the response in `A3` above, you can also find these newly added advanced methods in Table 2 and Line 820 of the revised manuscript.

---

> > > > > > ### Author Response · Authors · 2024-11-26
> > > > > > **Response to Reviewer F7Tj**
> > > > > >
> > > > > > Dear Reviewer F7Tj,
> > > > > >
> > > > > > Thank you for your thorough review of our work, BinaryDM, during the review stage. We have carefully considered your concerns during the rebuttal stage and made a revision to the relevant sections of the manuscript.
> > > > > >
> > > > > > We are looking forward you to reviewing our response and we are also willing to answer any further questions.
> > > > > >
> > > > > > Best regards,
> > > > > >
> > > > > > Authors of Paper 1841

---

### Author Response · Authors · 2024-11-21
**General Response**

We sincerely appreciate the positive feedback from the ACs and reviewers regarding BinaryDM. To provide a clearer understanding of our work, we summarize our main contributions and the strengths acknowledged by the reviewers as follows:

We propose a novel weight binarization approach for DMs, namely BinaryDM, pushing binarized DMs to be accurate and efficient by improving the representation and optimization. From the representation perspective, we propose the Evolvable-Basis Binarizer (EBB), enabling a smooth evolution of DMs from full-precision to accurately binarized models, thereby enhancing information representation during the initial training stages.  From the optimization perspective, we apply the Low-Rank Representation Mimicking (LRM) technique to aid the optimization of binarized DMs, alleviating the directional ambiguity inherent in the optimization process through fine-grained alignment.  Comprehensive experiments demonstrate that, as the first binarization method for diffusion models on generation, W1A4 BinaryDM achieves impressive OPs and model size savings, showcasing its substantial potential for edge deployment.

We have noted several concerns that reviewers have highlighted as particularly important:

- Rationale for applying EBB to the first and last six layers: This decision is based both on our understanding of its underlying principles and on straightforward, effective experimental validation. As explained in Section 3.2 of the manuscript, particularly in the “Location Selection” subsection, the motivation for EBB is to enhance the learning capability of binarized diffusion models (DMs) during the initial training stages by overcoming the structural limitations of binarization, ultimately improving the generative performance. However, such evolution comes at a cost, and suboptimal transitions may lead to undesirable outcomes, such as incomplete evolution. A natural approach is to target the most critical areas requiring enhanced learning capacity during the early stages. This allows us to boost information representation while avoiding the additional burden and potential risks of widespread adjustments. Therefore, we applied EBB to the first and last six layers, as these parts account for a small portion of the total parameters (less than 15%) and their proximity to the input and output makes them crucial to the generative performance of DMs. Based on the reviewers' suggestions, we have added more detailed theoretical analyses and experimental validations, thoroughly justifying our choice of EBB’s application range and its generalizability across different models and datasets.

- Comparison with other state-of-the-art methods: As requested by the reviewers, we have supplemented our comparisons with the most advanced general binarization methods, such as ReActNet and INSTA-BNN, as well as recent methods for binarizing diffusion models in super-resolution tasks, such as BI-DiffSR. Our findings show that BinaryDM still achieves the best generative performance. This addition builds upon our original manuscript, which already compared BinaryDM against state-of-the-art methods like EfficientDM and Q-DM, further providing a comprehensive and robust analysis of BinaryDM’s effectiveness and superiority.

For a detailed explanation, please refer to our responses to each reviewer. We have also incorporated the additional content into the revised manuscript, with updates marked in red.

---

### Meta-Review · Area_Chair_b8VP · 2024-12-19

**Metareview:**

The paper addresses the challenging problem of binarizing diffusion models. The proposed methods, EBB and LRM, are novel and tailored to address specific issues in binarizing DMs. A thorough experimental evaluation are given across multiple datasets and models. The results demonstrate that BinaryDM outperforms existing quantization methods under ultra-low bit-width settings.

**Additional Comments On Reviewer Discussion:**

All the reviewers lean to accept this paper.

---

### Decision · Program_Chairs · 2025-01-22

Accept (Poster)